# Potential Therapeutic Strategies in the Treatment of Metabolic-Associated Fatty Liver Disease

**DOI:** 10.3390/medicina59101789

**Published:** 2023-10-08

**Authors:** Aleksandra Bołdys, Łukasz Bułdak, Mateusz Maligłówka, Stanisław Surma, Bogusław Okopień

**Affiliations:** Department of Internal Medicine and Clinical Pharmacology, Medical University of Silesia, Medykow 18, 40-752 Katowice, Poland

**Keywords:** metabolic-associated fatty liver disease, incretins, diabetes, hyperlipidemia, obesity, pharmacotherapy

## Abstract

Metabolic-associated Fatty Liver Disease is one of the outstanding challenges in gastroenterology. The increasing incidence of the disease is undoubtedly connected with the ongoing obesity pandemic. The lack of specific symptoms in the early phases and the grave complications of the disease require an active approach to prompt diagnosis and treatment. Therapeutic lifestyle changes should be introduced in a great majority of patients; but, in many cases, the adherence is not satisfactory. There is a great need for an effective pharmacological therapy for Metabolic-Associated Fatty Liver Disease, especially before the onset of steatohepatitis. Currently, there are no specific recommendations on the selection of drugs to treat liver steatosis and prevent patients from progression toward more advanced stages (steatohepatitis, cirrhosis, and cancer). Therefore, in this Review, we provide data on the clinical efficacy of therapeutic interventions that might improve the course of Metabolic-Associated Fatty Liver Disease. These include the drugs used in the treatment of obesity and hyperlipidemias, as well as affecting the gut microbiota and endocrine system, and other experimental approaches, including functional foods. Finally, we provide advice on the selection of drugs for patients with concomitant Metabolic-Associated Fatty Liver Disease.

## 1. Introduction

Nonalcoholic fatty liver disease (NAFLD) is a disease with a complex etiology. Several years ago, the terminology was changed to Metabolic-Associated Fatty Liver Disease MAFLD because it better indicated the necessity for treating the risk factors of this disease. But in June 2023, during the EASL (European Association for the Study of the Liver) Congress in Vienna led by three major pan-national liver associations (ALEH—the Latin American Association for the Study of the Liver, AASLD—the American Association for the Study of Liver Diseases, and EASL) and the NAFLD Nomenclature Initiative, another transition was proposed. Steatotic liver disease (SLD) was selected as a comprehensive term to cover the diverse causes of steatosis. The proposed replacement for NAFLD is metabolic dysfunction-associated steatotic liver disease—MASLD. The modified definition includes at least one of the five cardiometabolic risk factors. Individuals without metabolic disease and no other known cause should be diagnosed with cryptogenic SLD. Outside the MASLD, a new clinical entity—MetALD (Metabolic Alcohol-Related Liver Disease)—was established for individuals consuming higher weekly alcohol amounts (140–350 g per week for females and 210–420 g per week for males). Metabolic dysfunction-associated steatohepatitis (MASH) is the replacement term for nonalcoholic steatohepatitis (NASH) [1]. However, owing to the practical approach in the manuscript, we refer to steatotic liver disease as MAFLD/NAFLD.

The development of MAFLD is treacherous. In the past, it has been considered as a benign disorder resulting from lipid accumulation in hepatocytes [2]. Currently, we perceive MAFLD as a risk factor of liver cirrhosis and a source of systemic complications [3]. The first stage in the development of MAFLD is liver steatosis, which results from the accumulation of lipids inside hepatocytes. At some point, owing to excess fat, leading to lipotoxicity and accompanied by unfavorable genetic factors, inflammation ensues. MASH, which is a following stage in the development of MAFLD, is characterized by the inflammatory response activation of various cells, including the macrophageal lineage and stellate cells, which contribute to the excessive intercellular matrix synthesis that is responsible for liver fibrosis [4]. In some patients, the advanced stage of liver fibrosis leads to cirrhosis and its complications, including hepatocellular carcinoma (HCC). Owing to widespread occurrence (around 25%), MAFLD might become a leading cause of liver fibrosis and HCC (Figure 1) [5].

There is an urgency to introduce effective therapies against MAFLD, especially at the early phase of its development. Currently, vitamin E and pioglitazone are recommended by international scientific organizations [7], but those drugs are used in more advanced stages of MAFLD (i.e., MASH), and the extent of the improvements in simple steatosis seems to be underwhelming. Therefore, there is significant interest in other therapeutic options that might be used to treat MAFLD, especially regarding the necessary pharmacological approach to other concomitant conditions in patients with features of metabolic syndrome (e.g., diabetes and hyperlipidemia). In this context, both the treatment and directions of development of MAFLD pharmacotherapy should be focused on seeking drugs that improve the courses of obesity, diabetes, insulin resistance, lipid disorders, hypertension, and hyperuricemia. Nevertheless, other potential therapeutic areas should also be explored (the endocrine system, microbiota, and functional foods).

One of the challenges in the introduction of new therapies is the proper assessment of their clinical efficacy. Despite having the best diagnostic yield, liver biopsies in an early stage of MAFLD (e.g., steatosis) are not recommended. The most accurate method of the noninvasive evaluation of liver steatosis is the MRI-proton density fat fraction (MRI-PDFF). However, the cost of the procedure and its limited availability mean that it is reserved nearly exclusively for clinical trials. There are other solutions to quantify steatosis. Ultrasound techniques (e.g., Fibroscan) are more accessible but less accurate than MRI. There are also several blood markers that can be used to estimate the extent or the prognosis of MAFLD (Fib-4, etc.). Nevertheless, most trials report simple liver function tests (LFTs—i.e., alanine aminotransferase—ALT, aspartate aminotransferase—AST, and gamma-glutamyl transferase—GGT). Herein, we provide an extensive, narrative review on the potential use of experimental pharmacological approaches to treat early stages of MAFLD—predominantly liver steatosis, focusing on the most substantial reported results (Figure 2).

## 2. Therapeutic Lifestyle Changes

The MAFLD incidence rises parallelly with the worldwide epidemic of obesity. The pathophysiological connection seems to be clear—excess fat intake, obesity, insulin resistance, and dyslipidemia [8]. Significant improvements in MAFLD may be achieved by implementing therapeutic lifestyle changes. Recent data suggest a positive correlation between the reduction in subcutaneous fat and the extent of liver steatosis [r = 0.42 (CI: 0.29–0.54)] [9]. This translates to improvements in LFTs. Dietary approaches have led to a decline in ALT activity [MD: (−4.48 IU/L)], and the effect was even more pronounced in patients who had also undertaken physical activity ALT [(−13.27) IU/L; 95% CI: (−21.39)—(−5.16)]. In those patients, an improvement in AST was also noted [(−7.02) IU/L; 95%: (−11.26)–(−2.78)] [5].

Recommendations for increased physical activity have been incorporated into the guidelines of most scientific societies dealing with MAFLD. AASLD [10] recommends that patients with NAFLD should be encouraged to increase their activity level to the maximum possible extent. Individualized prescriptive exercise recommendations may enhance sustainability and offer benefits independent of weight loss. EASL also recommends a progressive increase in aerobic exercise and resistance training [8].

One of the recently published systematic reviews [11] demonstrated that physical activity had a strong association with improvements in inflammation and reductions in steatohepatitis and fibrosis in experimental models. Furthermore, in human studies, both aerobic and resistance exercises were shown to reduce liver fat and improve insulin resistance and blood lipids, irrespective of weight loss, with aerobic exercises possibly being more effective than resistance exercises. This Review also shows that resistance training is more achievable for patients with NAFLD who have poor cardiorespiratory fitness (CRF). A meta-analysis by Wang [12] showed that physical activity was associated with minute reductions in LFTs: ALT [SMD = (−0.17 IU/L), 95% CI: (−0.30)–(−0.05)], AST [SMD = (−0.25 IU/L), 95% CI: (−0.38)–(−0.13)], and GGT [SMD = (−0.22 IU/L), 95% CI: (−0.36)–(−0.08)]. Similar findings were observed in the meta-analysis and meta-regression performed by Xiong et al. [13], which indicated that aerobic exercises in patients with NAFLD could significantly reduce the activities of ALT [WMD = (−6.14 IU/L), 95% CI: (−10.99)–(−1.29)] and AST [WMD = (−5.73 IU/L), 95% CI: (−9.08)–(−2.38)] and the BMI [WMD = (−0.85 kg/m^2^), 95% CI: (−1.19)–(−0.51)]. Additionally, resistance exercises could significantly reduce the AST activity [WMD = (−2.58 IU/L), 95% CI: (−4.79)–(−0.36)], while high-intensity interval training could significantly reduce the ALT activity [WMD = (−6.20 IU/L), 95% CI: (−9.34)–(−3.06)] in patients with NAFLD.

Therefore, currently, a 7–10% reduction in body weight accompanied by regular physical activity is universally recommended as a standard approach in overweight patients with MAFLD [14].

## 3. Drugs Affecting Carbohydrate Metabolism

Type 2 diabetes mellitus (DMT2) is one of the metabolic diseases that increases the risk for the development of MAFLD [15]. Crucial pathophysiological factors in DMT2, including hyperglycemia and insulin resistance, play a key role in the progression of MAFLD [16]. Drugs that affect carbohydrate metabolism have been considered to play a significant role in the pathogenesis of MAFLD. The development of MAFLD, as a consequence of the obesity and diabetes pandemics, has led to parallel advancements in the development of novel antidiabetic drugs that simultaneously exhibit weight-reducing effects, contributing to the reduction in MAFLD progression [17]. These drugs include SGLT-2 inhibitors (Sodium-Glucose Cotransporter-2)—leading to glucose excretion in the urine and lowering blood glucose levels; incretin analogs, such as GLP-1 (Glucagon-like Peptide-1)—stimulating insulin secretion, suppressing glucose release from the liver, and slowing down gastric emptying—and GLP-1 + GIP (Glucose-Dependent Insulinotropic Peptide)—having effects similar to those of GLP-1 and influencing blood glucose control; GLP-1 + glucagon (GCG)—potentially affecting glucose metabolism and aiding in diabetes management; as well as the most recent triple-incretin agonist, GLP-1/GIP/GCGR. The impacts of other antidiabetic medications have also been explored in the context of MAFLD: metformin—reducing glucose production in the liver and increasing tissue sensitivity to insulin, DPP4i (Dipeptidyl Peptidase-4 inhibitors)—increasing GLP-1 and GIP levels, ketohexokinase inhibitors—a promising new area of research that may influence glucose metabolism, and insulin sensitizers—glitazones. These medications have the potential to improve the condition of patients with carbohydrate disorders, including MAFLD. However, understanding the full extent of their impact and effectiveness in MAFLD therapy requires further clinical research.

### 3.1. Sodium/Glucose Cotransporter-2 Inhibitors

Sodium/glucose cotransporter-2 inhibitors (SGLT2is), also known as flozins, are drugs that are capable of limiting glucose reabsorption in renal proximal tubules. By inducing glucosuria, they subsequently lower serum glucose levels [18]. The first drug from the group of SGLT2is (canagliflozin) was accepted by the US Food and Drug Administration for the therapy of DMT2 in 2013. During the last 10 years, other effective flozins have been implemented for the management of DMT2 e.g., dapagliflozin, empagliflozin, and ertugliflozin [19].

Since the introduction of SGLT2is, their range of indications has vastly expanded. Drugs that originally were developed as antidiabetic medications appeared to have cardio- and nephroprotective effects. The results of clinical trials, including DAPA-HF, DAPA-CKD, EMPEROR-Reduced, EMPEROR-Preserved, and CREDENCE, allowed the application of selected SGLT2is in patients with heart failure (HF) or chronic kidney disease (CKD) even without concomitant DMT2 [20]. The beneficial effects of SGLT2is in cardiovascular and renal systems, for which no sufficiently effective drugs were available, led to the investigation of their potential for improving hepatic cell functioning.

Data derived from studies both on rodents and humans show the positive effects of therapy with SGLT2is on MAFLD. The cellular mechanisms responsible for the amelioration of liver functions are still not fully elucidated but are thought to be connected with the activation of autophagy, endoplasmic reticulum stress reduction, and anti-apoptotic, anti-inflammatory, and antioxidant effects [21,22,23,24].

In a randomized controlled trial (E-LIFT) in patients with DMT2 and NAFLD, empagliflozin reduced the serum ALT activity (−10.9 IU/L; *p* = 0.005) and reduced the amount of liver fat assessed with MRI-PDFF. The mean reduction in liver fat content was 4% in comparison with the control group (*p* < 0.0001) [11,25]. A systematic review of other randomized controlled trials (RCTs) confirmed that empagliflozin could ameliorate the plasma activity of AST [MD: (−3.10) IU/L; 95% CI: (−6.18)–(−0.02), *p* = 0.05]; liver stiffness, as assessed by an ultrasound examination [MD: (−0.49) kPa; 95% CI: (−0.93)–(−0.06), *p* = 0.03]; the homeostasis model assessment of insulin resistance (HOMA-IR) [MD: (−0.45); 95% CI: (−0.90)–0.00, *p* = 0.05]; and the body mass index (BMI) [MD: (−0.98) Kg/m^2^; 95% CI: (−1.87)–(−0.10), *p* = 0.03] [12]. There were no statistically significant differences in the reduction in other indicators of liver fibrosis e.g., neither the controlled attenuation parameter (CAP) nor the fibrosis-4 (FIB-4) index [26].

The data from meta-analyses concerning dapagliflozin are similar, with significant reductions in the ALT [WMD: (−6.62) IU/L; 95% CI: (−12.66)–(−0.58), *p* = 0.03] and AST [WMD: (−4.20) IU/L; 95% CI: (−7.92)–(−0.47), *p* = 0.03] levels, HOMA-IR [WMD: (−0.88); 95% CI: (−1.43)–(−0.33), *p* = 0.002], and BMI [WMD: (−1.33) Kg/m^2^; 95% CI: (−2.37)–(−0.28); *p* = 0.01] [9]. Dapagliflozin also reduced the CAP [MD = (−38.86) dB/m; 95% CI: (−73.39)–(−4.33)] [27] and liver fat content, as evaluated based on the liver attenuation index (LAI) in non-contrast computed tomography (CT) (5.8 ± 5.1 vs. 0.5 ± 6.1 Hounsfield units, *p* = 0.006) [28].

Improvements in liver function were also observed in patients with DMT2 and concomitant NAFLD treated with canagliflozin based on analyses from the CANVAS study. Clinically significant reductions in the ALT level or the normalization of ALT were more common in subjects undergoing active treatment (OR = 1.52; 95% CI: 1.4–1.65), *p* < 0.001). Furthermore, improvements, after 3 years of therapy, included an ALT level reduction (*p* < 0.001), as measured based on noninvasive tests of fibrosis, e.g., NAFLD fibrosis score (NFS) [between-group difference (−0.062); 95% CI: (−0.116)–(−0.008); *p* = 0.002] and the fibrotic nonalcoholic steatohepatitis index (FNI) [between-group difference (−0.054); 95% CI: (−0.067)–((−0.041); *p* < 0.001] [29]. Moreover, SGLT2is appear to prevent the progression of liver diseases and inflammation [30]. Although it seems that the beneficial class effect may be present in SGLT2is, the currently available data show that dapagliflozin improves not only LFTs but also the fat content, as assessed based on the CAP.

### 3.2. Agonists of Glucagon-like Peptide-1 Receptors

GLP-1 receptor agonists (GLP-1Ras) have shown substantial advantages in managing diabetes and obesity. Those properties are reflected by their action on pancreatic islets, beta cells, and the central nervous system [31,32,33]. Despite these benefits, the response rates to GLP-1Ras, as well as to other pharmaceutical treatments for liver steatosis in NAFLD, have not surpassed 50%. Research data suggest that GLP-1Ras may have beneficial effects on MAFLD by improving insulin sensitivity, reducing liver fat accumulation, and potentially exerting anti-inflammatory effects. These drugs have shown promise in reducing LFTs. They also could have an influence on the liver histology. Liraglutide is the most extensively investigated GLP-1 receptor agonist (GLP-1 RA) in the context of NAFLD treatment. Several trials have evaluated its effectiveness in these individuals, yielding positive results. The multicenter, double-blinded, randomized, placebo-controlled phase 2 trial showed the benefits of a GLP-1 analog (liraglutide) [34] in patients with MASH. A 48-week treatment with liraglutide at a dose of 1.8 mg (26 patients) resulted in histological improvement, without exacerbating fibrosis, when compared to the placebo group (also 26 patients) (RR = 4.3; 95% CI: 1.0−17.7, *p* = 0.0190). The primary histological outcome endpoint was an improvement in liver histology, which was defined as the resolution of steatohepatitis characterized by the disappearance of hepatocyte ballooning. The secondary endpoint included changes in the total NAFLD activity score (steatosis, hepatocyte ballooning, lobular inflammation, and fibrosis stages) as well as the serum liver enzyme concentration. The differences in the lobular inflammation and overall NAFLD activity scores were not statistically significant between the two groups.

Semaglutide is being assessed for reducing liver steatosis in patients with NAFLD who are undergoing antiretroviral therapy [35]. Additionally, baseline parameters predicting the clinical response in NAFLD during semaglutide treatment [36] and the applicability of semaglutide (oral or subcutaneous form) as an effective measure in improving NAFLD in patients with obesity and/or type 2 diabetes mellitus [37] are also being investigated. Orforglipron (also known as LY3502970 or OWL833), a new oral GLP-1 analog, is currently under investigation in phase 1 and 2 clinical trials [38,39] in patients with diabetes or obesity and who are overweight and have weight-related comorbidities (including MAFLD) [40].

Most studies investigating the efficacy of GLP-1 analogs are focused on their application in obesity and MASH treatments, thereby leaving a substantial area for exploration concerning their effectiveness in earlier stages of NAFLD, as a preventive measure in MASH development. Currently, the AASLD guidelines [41] strongly recommend the use of semaglutide, 2.4 mg/week (according to the strongest supporting evidence), or liraglutide, 3 mg/day, for chronic weight management for individuals with a BMI of ≥27 kg/m^2^ and either NAFLD or MASH, supporting their role in the treatment of liver disease.

The strength of GLP-1 agonists lies in their effectiveness in reducing body weight, which constitutes a fundamental element of MAFLD treatment. The limitations of the presented studies include, among others, small patient group sizes (as observed in the LEAN study on liraglutide) as well as the assessed parameters. These studies primarily focus on the efficacy of these drugs for treating MASH, including the reduction in fibrosis. Further research is needed to provide robust evidence of liver steatosis reduction and factors involved in the progression from MAFLD to MASH (including oxidative stress factors).

### 3.3. Dual Agonists of Incretin Receptors (Glucagon-like Peptide-1 and Glucose-Dependent Insulinotropic Polypeptide)

Although GLP-1 is thought to affect the liver by reducing body weight, the mechanism of action of the dual GLP-1 and GIP receptor agonists is still not fully understood. It could result from GLP-1 agonism enhanced by the GIP action, but the exact mechanism of action is still unknown as there is still a lack of human data. One of the explanations suggests a potential impact on liver cells through an as-yet-unknown mechanism. As presented in their recent study on tirzepatide, Muller et al. indicated that signaling via the GIP receptor in human pancreatic islets is crucial for stimulating insulin secretion, unlike in mice where the action of tirzepatide is primarily associated with activity through GLP-1 [42]. The tirzepatide—dual glucose-dependent insulinotropic polypeptide and glucagon-like peptide-1 receptor—agonist demonstrated substantially greater glucose control and weight loss compared with the selective GLP-1RA—dulaglutide [43]. Tirzepatide is currently under evaluation in the SYNERGY-NASH trial [44] in MASH patients without cirrhosis. In a post hoc analysis, patients who received tirzepatide experienced improvements in MASH-related biomarkers, such as ALT, CK-18 (cytokeratin-18), and Pro-C3 (N-terminal type III collagen propeptide), as well as an increase in the adiponectin level [45].

In the field of liver steatosis in early NAFLD, tirzepatide has been investigated in the SURPASS-3 MRI study [46] (a substudy of a randomized, open-label, parallel-group, phase 3 SURPASS-3 trial) where the primary focus was on evaluating the change in the liver fat content (LFC), as assessed using MRI-PDFF. This analysis involved combined data from tirzepatide (10 mg and 15 mg groups) and compared them with data from study participants treated with insulin degludec. After 52 weeks of treatment, there was a statistically significant and greater absolute reduction in liver fat content for the combined tirzepatide groups (reduction of 8.09%, SE 0.57) compared to the insulin degludec group (reduction of 3.38%, SE 0.83). The reduction in LFC was significantly associated (*p* ≤ 0.0006) with several factors. including the baseline LFC (correlation coefficient R = −0.71), decreases in visceral adipose tissue (VAT) (R = 0.29), reductions in alanine aminotransferase (ALT) levels (R = 0.33), and decreases in body weight (R = 0.34) within the tirzepatide groups. Those preliminary results suggest the potential utilization of tirzepatide in the treatment of NAFLD. As a relatively new drug that shows significant effectiveness in weight reduction, it may become the focus of many subsequent studies assessing its efficacy in the treatment of MAFLD once the drug is more widely introduced to the market. The initial results appear promising, and further research may potentially demonstrate the significance of tirzepatide’s agonistic action on the GIP receptor.

### 3.4. Dual Agonists of Incretin Receptors (Glucagon-like Peptide-1 and Glucagon)

The simultaneous activation of GLP-1 and glucagon receptors prevents the hyperglycemic response commonly associated with glucagon while also enhancing its catabolic effects and significantly amplifying hepatic glycolysis, glycogenolysis, and lipolysis. GLP-1 activation has been correlated with weight reduction, anorexigenic properties, and hypoglycemic effects, while the activation of GCGR is believed to mainly contribute to a reduction in hepatic steatosis and enhancement in mitochondrial respiration; the dual agonist of GLP1/GCG receptors are believed to improve the course of NAFLD and are currently being investigated for this indication.

The recently published results of a phase 2a active-comparator-controlled study [47] on efinopegdutide (NCT: 04944992) demonstrated that in 145 randomized patients with NAFLD, 24 weeks of treatment with a weekly dose of efinopegdutide at 10 mg (72 patients) resulted in a substantial reduction (*p* < 0.001) in liver fat content (LFC), as measured using magnetic resonance imaging (LFC reduction of 72.7% [90% CI: 66.8–78.7]) when compared to a weekly dose of semaglutide at 1 mg (73 patients, LFC reduction of 42.3% [90% CI: 36.5–48.1]). In a phase 2b study involving more than 800 overweight diabetic subjects with inadequate blood glucose control, another GLP1R/GCGR agonist, cotadutide, administered at dosages of 100–300 μg/d, yielded significant improvements in liver enzyme levels and indicators of liver fibrosis. In this study, cotadutide achieved noteworthy reductions in HbA1c and body weight at both 14 and 54 weeks compared to the placebo group (all *p* < 0.001). Additionally, 300 μg of cotadutide demonstrated improvements in the lipid profile, AST and ALT levels, propeptide of the type III collagen level, fibrosis-4 index, and nonalcoholic fatty liver disease fibrosis score compared to the placebo group, but this effect was not observed with liraglutide. [48]. Currently, another ongoing study is assessing the safety and efficacy of cotadutide in participants with non-cirrhotic MASH and fibrosis [49]. According to a review article on clinical trials of MASH published in July 2023 [50], cotadutide studied in 74 patients demonstrated a moderate impact on the hepatic fat fraction, ALT, and AST after 19 weeks in the cotadutide cohort compared to the placebo group.

Other dual GLP1/GCGR agonists intended for use in NAFLD treatment are being investigated. Pemvidutide is under development as a treatment for obesity and MASH. According to a clinical-stage biopharmaceutical company announcement dated March 21, 2023 and the aforementioned review article [50], a 12-week phase 1b trial for NAFLD patients showed that 65%, 94.4%, and 85% of patients treated with pemvidutide (1.2 mg, 1.8 mg, and 2.4 mg, respectively) achieved a ≥30% relative reduction in hepatic fat on MRI-PDFF in comparison with 4.2% in the placebo group. However, to date, there have been no other publications available from these two completed studies investigating pemvidutide and assessing its safety and effect on the hepatic fat fraction in subjects with NAFLD after 12 and 24 weeks of treatment (MOMENTUM Obesity trials) [51,52].

Another candidate that is potentially applicable to NAFLD treatment [53] is a drug combination (HM14320) consisting of HM15136 (a glucagon analog) and efpeglenatide (a GLP-1 analog), but there are no published studies regarding the combination’s effectiveness. Moreover, the preclinical studies on obesity and MASH were discontinued in 2022 [54].

Furthermore, survodutide (BI 456906), a dual agonist of the GLP-1 and glucagon receptors, is being evaluated. Clinical trials compare its effectiveness mostly with semaglutide (obesity). Currently, there are no ongoing studies on survodutide in NAFLD treatment; the only ongoing study pertains to MASH [55]. Also, research on the effectiveness of danuglipron in NAFLD treatment was discontinued because the sponsor decided to seek its application only in patients with obesity and DMT2 (phase 2 clinical trial) and no active liver diseases [56].

Apparently, most studies related to this group of drugs in MAFLD treatment conclude at preclinical investigations or early stages of clinical trials. The future application of these drugs in the treatment of MAFLD without concurrent diabetes currently appears unlikely. Furthermore, the question of the extent to which the activation of the glucagon receptor influences the liver fat reduction and whether actions at the mitochondrial level may have a significant impact on this reduction remains to be settled.

### 3.5. Triple Incretin Receptor Agonists

The triple GLP1R/GCGR/GIPR agonist, efocipegtrutide (HM15211), distinguished by its prolonged duration of action, which is attributed to a non-peptidyl flexible linker, is being investigated in a phase 2 clinical trial for MASH (217 participants, aged 18–70 years, across multiple US sites with an expected completion date in November 2025) [57], and a phase 1 trial to evaluate the safety, tolerability, pharmacokinetics, and pharmacodynamics of multiple doses of HM15211 in obese subjects with NAFLD has been reported (66 participants, aged 18–65 years) [58]. However, at present, no peer-reviewed articles presenting the results of studies involving efocipegtrutide are available.

Retatrutide (LY3437943), another triple agonist, is presently in development for the treatment of DMT2, obesity, and nonalcoholic fatty liver disease. Data emerging from the phase 2 study [59], specifically within the NAFLD subgroup, indicate the potential for reversing initial liver disease stages and mitigating obesity-related coexisting conditions. The results revealed that within the NAFLD population, liver fat normalization was achieved in 90% of cases subsequent to treatment with the two most elevated doses. [60].

The results of a newly published network meta-analysis by Kongmalai et al. [61], involving 2252 patients from 31 randomized controlled trials, demonstrated that the addition of GLP-1 agonists to the standard of care in NAFLD patients’ treatment led to a significant reduction in intrahepatic steatosis (IHS) when compared to the standard treatment alone. The effect size indicated a reduction of −3.93% (95% CI: −6.54% to −1.33%) in steatosis. The cumulative ranking curve (SUCRA) analysis revealed that GLP-1 receptor agonists had the highest probability (SUCRA 88.5%) of reducing IHS. GLP-1 receptor agonists were also found to be the most effective in reducing liver enzyme levels, specifically AST [MD (−5.04) IU/L; 95% CI: (−8.46)–(−1.62)], ALT (MD: (−9.84) IU/L; 95% CI: (−16.84)–(2.85)], and GGT [MD: (−15.53) IU/L; 95% CI: (−22.09)–(−8.97)] in comparison to the standard of care, but they were more likely to be associated with adverse events compared to other interventions.

Triple incretin agonists might become useful in the treatment of NAFLD; but, currently, the development of clinical programs is in their early phases. Therefore, we must wait for more solid evidence to provide clinical recommendations.

### 3.6. Dipeptidyl Peptidase-4 Inhibitors

Dipeptidyl peptidase-4 (DPP-4) inhibitors, which increase levels of endogenous incretins, have also been investigated for their potential application in MAFLD treatment. They might help to improve insulin resistance and glucose metabolism, which are central factors in MAFLD development. A surface under the cumulative ranking curve analysis (SUCRA) revealed that DPP-4 inhibitors had the second highest (following GLP-1 agonists) probability (SUCRA 69.6%) of reducing intrahepatic steatosis, followed by pioglitazone (SUCRA 62.2%) [61]. However, further studies seem to be necessary to determine the full extent of their effects on MAFLD progression.

DPP-4 inhibitors constitute a category of indirect incretin mimetics as they impede the enzymatic breakdown of GLP-1, GIP, and oxyntomodulin; but, in the available guidelines or recommendations [41,62], there are still few studies focusing on hard end-points (inflammatory markers or fibrosis) regarding DPP-4 inhibitors in the treatment of NAFLD. That was demonstrated by dos Santos et al. [63], who indicated the overall poor quality of the studies and heterogeneity of the analyzed population. Nevertheless, several limited-scale clinical trials have investigated the potential efficacy of DPP-4 inhibitors in NAFLD treatment, both with and without accompanying DMT2.

Among these trials, sitagliptin (100 mg/d) displayed effectiveness against hepatic steatosis and the hepatic collagen content regardless of DMT2, as demonstrated in a 1-year open-label randomized controlled trial [64]. Vildagliptin in DMT2 patients [65] and non-diabetic patients [66], omarigliptin [67], and teneligliptin (only in MASH) [68] exhibited improvements in liver function and certain noninvasive markers of NAFLD. The preliminary data for saxagliptin patients with DMT2 and concomitant NAFLD demonstrated that saxagliptin could attenuate insulin resistance and inflammatory injury by the downregulation of the hepatic and soluble form of DPP-4 and, as a result, reduce the degree of steatosis [69,70].

Alogliptin showed only moderate efficacy against NAFLD (as measured based on the NAFIC score—NASH, ferritin, insulin, and collagen 7S score) over a 12-month treatment period in patients with DMT2 and NAFLD [71]. Other DPP-4 inhibitors, like evogliptin, anagliptin, trelagliptin, gemigliptin, and linagliptin, have demonstrated positive effects in experimental rodent models, and their clinical utility remains to be explored in forthcoming trials, as indicated by Pirkhodko [53] in her review.

### 3.7. Glitazones

The AACE (American Association of Clinical Endocrinology) [41] and AASLD [10] clinical practice guidelines endorse the utilization of pioglitazone, a selective PPARγ (Peroxisome proliferator-activated receptor gamma) agonist, in cases of diagnosed MASH. This recommendation stems from the studies in which, irrespective of the presence of DMT2, pioglitazone reduced the degree of liver steatosis and improved disease activity indicators [72,73,74,75,76]. In the field of NAFLD, a one-year treatment with pioglitazone (compared to sulfonylureas) at a low dosage significantly (*p* < 0.05) improved liver steatosis and inflammation and systemic and adipose-tissue insulin resistance in patients with DMT2 [77]. Surrogate markers of NAFLD were improved: the liver fat equation decreased by (−1.76) ± 3.84 (*p* < 0.05); the hepatic steatosis index by (−1.35) ± 2.78 (*p* < 0.05); and the index of MASH by (−9.75) ± 43 (*p* < 0.05). Other thiazolidinediones considered in the treatment of NAFLD–rosiglitazone [78] and lobeglitazone (available only in Republic of Korea)—do not have a substantial number of studies confirming their effectiveness. So far, there has been only one paper that has emerged after the phase 4 clinical trial [79] of DMT2 and NAFLD patients treated with lobeglitazone (43 who completed study), showing a significant decrease in CAP values (313.4 dB/m at baseline vs. 297.8 dB/m at 24 weeks; *p* = 0.016), regardless of the glycemic control [80].

### 3.8. Ketohexokinase Inhibitors

KHK inhibitors (KHKis) are novel, promising compounds in the treatment of NAFLD. Fructose (C_6_H_12_O_6_) is a keto-hexose (ketose-hexose) isomer of glucose, and it plays a significant role in the development of NAFLD. KHK is an enzyme that is responsible for the initial and critical step in fructose metabolism, which is believed to increase the intrahepatic lipid (IHL) content. The pharmacological inhibition of KHK has led to a reduction in the IHL content among individuals with NAFLD; however, the full scope of KHK inhibition remains to be elucidated. The safety and efficacy of PF-06835919 (a KHKi developed by Pfizer) have been assessed through several studies. KHK inhibition has been previously investigated in a small phase 2 trial involving individuals with NAFLD. PF-06835919 exhibited a significant reduction in liver LFC after 6 weeks of treatment compared to the placebo group. There were data indicating that participants receiving PF-06835919 at the 300 mg dose demonstrated significant reductions in LFC compared to the placebo group, with a difference of −18.73% (*p* = 0.04) [81] Another phase 2 study conducted on a group of patients with NAFLD and DMT2 demonstrated that at week 16, the least-squares mean (90% CI) percentage changes from baseline in LFC using MRI-PDFF were as follows: (−5.26%) [(−12.86%)–2.99%] in the placebo group, (−17.05%) [(−24.01%)–(−9.46%)] in the 150 mg dose PF-06835919 group, and (−19.13%) [(−25.51%)–(−12.20%)] in the 300 mg dose PF-06835919 group. Notably, the 300 mg dose PF-06835919 group exhibited a statistically significant reduction in LFC compared to the placebo group (*p* = 0.0288) [82]. The currently ongoing study [83] aims to explore additional health effects resulting from KHK inhibition with PF-0683591 in NAFLD patients without DMT2.

### 3.9. Metformin

Metformin, one of the basic drugs for treating DMT2, which enhances insulin sensitivity in the liver and muscles, has also been broadly studied for NAFLD treatment. According to AASLD recommendations [10] metformin (as well as acarbose—an alpha glucosidase inhibitor) should not be used for the treatment of steatohepatitis (as there is no observed benefit for hepatocyte necrosis or inflammation). However, it may be continued if needed for the treatment of hyperglycemia in people with DMT2 and NAFLD or MASH [84]. As several meta-analyses of paired-biopsy studies involving individuals with MASH have revealed, there has been limited clinical evidence indicating benefits in terms of disease activity or liver fibrosis [85,86] Preliminary studies indicated that a moderate effect was observed, mainly targeting hepatic steatosis and linked to weight loss [87,88]. Nevertheless, a meta-analysis of metformin trials revealed that the aggregated liver histologic scores for steatosis, ballooning, and fibrosis did not exhibit significant improvements. Furthermore, lobular inflammation significantly worsened (weighted mean increase, 0.21; 95% CI: 0.11–0.31; *p* < 0.0001), which is in line with findings from other systematic reviews and meta-analyses.

In summary, medications affecting carbohydrate metabolism are well-established within the AASLD guidelines [10]. Those include pioglitazone, liraglutide, semaglutide, tirzepatide, and SGLT-2 inhibitors owing to their proven histological benefits in patients with biopsy-confirmed MASH and concurrent cardiac benefits. However, according to AASLD, none of these medications are officially recommended for the treatment of MASH, but they can be employed in carefully chosen individuals with MASH who have comorbidities, such as diabetes or obesity.

## 4. Drugs Affecting Lipid Metabolism

### 4.1. Core Cholesterol-Lowering Therapy (Statins or/and Ezetimibe)

Hypercholesterolemia may lead to the progression of liver damage in MAFLD [89]. Statins play key roles in the therapy of hyperlipidemias [90]. However, many hepatotoxic properties have been attributed to their action. The cautious approach is reflected in the recommendations on the surveillance of statin therapy, which include routine checkups of ALT during the treatment of hyperlipidemia [91]. These drugs should not be used or should be withdrawn when the ALT level exceeds 3 times the upper limit of the reference range. Liver steatosis is generally associated with only a mild elevation in ALT levels. Interestingly, there have been reports on the reduction in LFTs during statin therapy [92]. This might stem from a direct reduction in intracellular cholesterol load as well as the pleiotropic actions of statins [93].

One of the first clinical trials focusing on the impact of a 20 mg dose of atorvastatin LFTs in patients with MAFLD was performed by Athyros et al. [94]. After 54 weeks of treatment, there were no statistically significant impacts of atorvastatin on ALT, AST, and ALP (alkaline phosphate) activities, which might be attributed to a relatively small sample size (63 patients). In a larger trial (GRAECE) subjects with mildly elevated aminotransferases (up to 3× above the reference range) were also treated with the statin. During the study a gradual reductions in the ALT, AST, and GGT activities were observed. At the end of observation (3 years), the ALT activity dropped from 57 ± 8 to 37 ± 6 IU/L (*p* < 0.0001); the AST and GGT activities were affected to a similar extent. No significant liver toxicity was reported for the statins. This study showed that surrogate markers of MAFLD might be improved during statin treatment [95]. In a recently published paper (ESSENTIAL study) on rosuvastatin (5 mg per day) in NAFLD treatment, there was no change in the liver function tests during the 24-week observation [96]. However, a significant reduction in fat accumulation was noted using MRI-PDFF (15.0 ± 7.3 vs. 12.4 ± 7.4%; *p* = 0.003). Interestingly, the reduction in the liver fat content was even greater in patients subjected to combined treatment with 5 mg of rosuvastatin and 10 mg of ezetimibe (absolute mean difference = 3.2%; *p* = 0.02). On the other hand, the significant lipid-lowering potency (LDL concentration reduction from 171 ± 24 to 95 ± 36 mg/dl; *p* < 0.001) of the combined therapy consisting of 10 mg of rosuvastatin and 5 mg of ezetimibe did not lead to any significant changes in aminotransferase activities (ALT: 71 ± 37 vs. 66 ± 37 IU/L; *p* = 0.051, AST: 47 ± 35 vs. 37 ± 17 IU/L; *p* = 0.215) in subjects with NAFLD during the 48-week treatment period [97]. The CAP score was also not affected during the trial (327 ± 76 vs. 307 ± 43 dB/m; *p* = 0.302). Significant improvements were noted only in subjects that, in addition to pharmacotherapy, introduced a therapeutic lifestyle change. Finally, the use of ezetimibe as a monotherapy improved the NAFLD activity score [SMD, (−0.30); 95% CI: (−0.57)–(−0.03)] but lacked an impact on the liver fat content, as assessed using MRI-PDFF or a liver biopsy [SMD (−1.01); 95% CI: (−2.03)–0.01] [98].

According to recently aggregated data, the impact of statin in patients with NAFLD seems to significantly reduce the ALT (by 35.41%), AST (by 31.78%), and GGT (by 25.57%) [99]. The impact on the LFT is simultaneous with lipid profile improvements. But it is conceivable that the pleiotropic effects of statins (e.g., anti-inflammatory and antioxidant) may have an additional important contribution. Such an impact might be, to some extent, responsible for the reduced propensity of hepatic cell carcinoma in NAFLD patients who are on statin therapy (OR = 0.59; 95% CI: 0.39−0.89) [100]. The available data support the notion that patients with MAFLD should not be excluded from statin therapy owing to a slightly elevated LFT prior to the initiation of the therapy. In fact, improvements in LFTs may be expected during prolonged treatments. Both atorvastatin and rosuvastatin seem to be favorable in NAFLD treatment.

### 4.2. PCSK9 Inhibitors

Although statins and ezetimibe have been used for several decades, PCSK9 inhibitors (PCSK9i) are relatively new in the therapy of hypercholesterolemia. No direct link has been shown between the circulating PCSK9 level and markers of NAFLD [101]. However, there are specific mutations in the PCSK9 gene that were connected to liver steatosis (e.g., c.946 G.T and p. Gly316Cys) [102], but there has been no connection with the advancement toward liver fibrosis [103]. Interestingly, specific loss-of-function mutations (e.g., rs11591147 R46L) in the PCSK9 gene seem to be protective against the development of MASH [104]. During the therapy with PCSK9i, patients with familial hypercholesterolemia, seemed to benefit beyond the lipid-lowering effects of the therapy. In a subset of patients with NAFLD and a low TG (triglyceride)/HDL (high density lipoprotein) ratio, a 6-month therapy with PCSK9i significantly reduced the expression of the surrogate markers of liver steatosis; the triglyceride-glucose index was reduced by 7.5% (*p* < 0.05) and the hepatic steatosis index by 8.4% (*p* < 0.05) [105]. Real-world data from a single site showed long-lasting (mean duration of therapy 23.69 ± 11.18 months) resolution of the radiologic features of liver steatosis in 8 out of 11 patients, which was accompanied by a significant ALT reduction (21.83 ± 11.89 vs. 17.69 ± 8.00 IU/L; *p* = 0.042) [106].

### 4.3. Peroxisome Proliferator-Activated Receptors Alpha Agonists

Fibrates belong to a family of PPAR alpha-receptor agonists. The main indication for these drugs is the treatment of hypertriglyceridemia. Their use since the introduction of the therapy has necessitated the periodical evaluation of aminotransferases owing to potential liver toxicity. However, studies in patients with MAFLD, who often have slightly elevated ALT levels, provide interesting input. It seems that fenofibrate at a dose of 300 mg per day for 24 weeks in an RCT reduced LFTs, including ALT (114.6 ± 8.5 vs. 57.3 ± 7.8 IU/L, *p* < 0.05). Additionally, indices of potential prevention in the progression toward more advanced stages of NAFLD were represented by a reduced TGF (transforming growth factor) beta level (15.2 ± 5.1 vs. 8.1 ± 3.2 ng/mL; *p* < 0.05) and reduced liver stiffness (12.95 ± 5.1 vs. 9.8 ± 3.7 kPa; *p* < 0.05) [107]. Similar results were noted in another RCT (randomized controlled trial) with a 200 mg dose of fenofibrate. The ALT level, at the end of study, was reduced from 78 ± 11 to 53 ± 26 IU/L (*p* = 0.002), and the extent of the reduction was similar to that achieved with the use of pioglitazone [108]. Contrary to the positive effects on the surrogate markers of NAFLD, a 12-week treatment of patients with diagnosed hypertriglyceridemia and an MRI-PDFF above 5.5% showed no influence on the ALT level and a slight increase in the AST level compared to the placebo group. Additionally, during treatment with fenofibrate, a significant increase in liver fat volume was noted (23%), but it was still comparable to the placebo arm [109]. Those results were somewhat unexpected, while the previous results showed a trend toward reduced lipid accumulation in the liver during fibrate therapy. Currently, fenofibrate is considered as a potential add-on to the novel therapies for MASH: (1) acetyl-CoA carboxylase (ACC) inhibitors (e.g., firsocostat), which leads to an elevation in TG. It seems that fenofibrate not only prevents the rise in TG but also reduces the ALT (−37.3%; *p* < 0.05), GGT (−34%; *p* < 0.05) and ALP (−14.1%; *p* < 0.001) levels compared to the baseline values [110]; and (2) farnesoid X receptor (FXR) agonists (e.g., cilofexor) [110].

### 4.4. Selective Peroxisome Proliferator-Activated Receptor Modulators

Fibrates, which are described above and indicated in the therapy of hypertriglyceridemia, seem to have some beneficial impact on the course of NAFLD. A similar mechanism of action, which is based on selective PPAR alpha modulation (SPPARM), is used by a novel compound—pemafibrate. The drug, itself, possesses triglyceride-lowering properties [111], but it excels at improving NAFLD. Early small-scale studies showed a remarkable improvement in the reduction of ALT from 57.5 ± 8.8 to 30.3 ±5.8 IU/L (*p* < 0.01) and GGT from 63.9 ± 10.3 to 32.8 ± 6.6 IU/L (*p* < 0.01) during a 6-month treatment [112]. Even shorter studies with pemafibrate show substantial improvements in LFT, as shown by Seko et al. [113]. The ALT level was reduced from 75.1 IU/L to 43.6 IU/L (*p* = 0.001) at week 12; but, even in the intermittent analysis at 4 weeks of treatment, ALT showed a significant reduction in LFTs compared to the baseline values. However, there was no significant change in the more-solid NAFLD endpoints (FIB-4 index: 1.89 ± 1.17 vs. 1.95 ± 1.24; *p* = 0.351, CAP: 329.1 ± 35.6 vs. 314.6 ± 51.6 dB/m; *p* = 0.329, LSM [liver stiffness measurements]: 10.5 ± 8.8 vs. 9.2 ± 7.6 kPa *p* = 0.080). Those data underline the necessity of prolonged observations to assess drug efficacy. Similar results were obtained by Shinozaki et al. [114]. A 3-month therapy with pemafibrate for 38 NAFLD patients resulted in significant reduction in ALT (63.9 ± 3.6 vs. 41.6 ± 3.6; *p* < 0.001), which was accompanied by improvements in the NAFLD fibrosis score [(−2.27 ± 0.18) vs. (−2.38 ± 0.18); *p* = 0.009], but no change in the FIB-4 index was noted (1.51 ± 0.16 vs. 1.47 ± 0.12; *p* = 0.500). A longer, 6-month, observation also led to improvements in aminotransferases, which were accompanied by a reduction in the FIB-4 score [2.26 (1.07–3.12) vs. 2.08 (0.97–2.67); *p* = 0.041]. Nevertheless, CAP and LSM remained unchanged [115]. A 12-month therapy with pemafibrate also remained effective in terms of ALT and AST [116]. The above-mentioned studies were retrospective, but the results were also favorable in the phase 2 RCT for a 0.2 mg dose of pemafibrate in NAFLD patients. During the 72-week study, a continuous reduction in ALT was reported (−33.6%) [(−46.5)–(−20.7); *p* < 0.0001] [117]. The liver fat content did not change significantly (−5.1%) [(−15.8)–5.6]; *p* = 0.35], whereas the liver stiffness, as assessed using MRI, was reduced by 6.2% (0.8–11.5; *p* = 0.024). There were no changes in the CAP and LSM in subjects with NAFLD treated for 48 weeks [118,119], but a reduction in liver stiffness was observed [1.45 m/s at baseline vs. 1.32 m/s at week 48 (*p* < 0.001)] [120]. Although most patients with MAFLD are overweight or obese, it seems that a greater benefit from the therapy might be observed in patients with a BMI of <25 kg/m^2^ [121].

### 4.5. Multiple Peroxisome Proliferator-Activated Receptor Agonists

Saroglitazar is used for the treatment of diabetic dyslipidemia. Owing to its mechanism of action, it improves both the lipid profile (PPAR alpha agonist) and glucose metabolism (PPAR gamma agonist). Additionally, since the beginning of its use, it has been noted that saroglitazar might improve the course of comorbid MAFLD. A case series of 10 patients treated with saroglitazar at 4 mg per day and followed-up for nine months resulted in significant reductions in ALT (64.7 ± 15.56 vs. 46.2 ± 12.6 IU/l; *p* = 0.0058) and AST (43.4 ± 10.48 vs. 35.4 ± 6.59 IU/l; *p* = 0.0321) activities. Those findings were accompanied by a significant reduction in SWV—Shear-Wave Velocity (1.837 ± 0.0691 vs. 1.645 ± 0.0844; *p* = 0.0004), which might be considered as a surrogate marker of fibrosis [122]. The limitation of this observation was its retrospective nature without a control group. In a larger cohort of patients with NAFLD, although still without control group, saroglitazar also improved LFT, including ALT (56.47 ± 15.17 vs. 42.3 ± 11.26 IU/l; *p* < 0.0001), AST (48.57 ± 13.15 vs. 36.63 ± 8.14 IU/l; *p* < 0.0001), and GGT (54.97 ± 9.52 vs. 45.33 ± 5.94 IU/l; *p* < 0.0001) [123]. Additionally, a Fibroscan assessment showed an impressive reduction in liver stiffness (13.93 ± 2.87 vs. 8.50 ± 1.86 kPa). Unfortunately, no data on the CAP level were included in the report. Consistently, positive results have also been shown in an even larger group of 107 patients during 24 weeks of therapy with saroglitazar. In addition to improvements in LFTs [ALT: 94 (47–122) vs. 39 (31–49) IU/L; *p*< 0.0001, AST: 89 (43–114) vs. 37 (30–47) U/L; *p*< 0.0001] and LSM: 8.4 (7.1–9.3) vs. 7.5 (6.4–8.4) kPa; *p* = 0.0261], a significant reduction in liver steatosis was also observed [CAP 335 (281–392) vs. 256 (212–299) dB/m; *p* = 0.0076] [124]. Comparable results have been reported by other researchers [125,126]. The most recent results obtained from the real-world use of saroglitazar (4 mg per day) in patients with MAFLD provide further promising data for MAFLD patients, even those progressing toward cirrhosis [MD AST (IU/L): 40.78%, *p* < 0.001; ALT (IU/L): 52.21%, *p* < 0.001; LSM (kPa): 22.11%, *p* < 0.00; CAP (dB/m): 328 (46): 14.18%, *p* < 0.00]. However, there seems to be paucity in the results of placebo-controlled double-blinded trials. Although data in such studies are available for LFT [127], we are still awaiting outcomes for CAP and LSM.

Other dual agonists of PPAR (i.e., alpha and delta—elafibranor) and pan PPAR agonists (i.e., alpha, gamma, and delta—lanifibranor) show promise in the treatment of MASH rather than steatosis [128,129].

### 4.6. Acetyl-CoA Carboxylase Inhibitors

Cellular lipid overload is a consequence of both exogenous and endogenous lipid excesses [130]. Experimental data strongly suggest that patients with MAFLD experience elevated (over 3-fold) de novo fatty-acid synthesis [131]. Therefore, endogenous synthesis seems to be an important target for the pharmacological approach to MAFLD treatment. One of the options is the inhibition of hepatocyte-specific Acetyl-CoA carboxylases (ACC). Initial experiments on MK-4074, an ACC1 and ACC2 inhibitor, led to a remarkable reduction in the liver’s fatty-acid content, which reached 36% (8.6% in placebo group) after a 4-week therapy [132]. Unfortunately, probably owing to the concurrent inhibition of the PUFA-3 elongation, a rise in the TG level was noted (170 vs. 325 mg/dL). Further, phase II clinical trials on another ACC inhibitor (firsocostat) showed dose-dependent reductions in liver steatosis, as assessed based on MRI-PDFF, reaching 28.9% (vs. 8.4%; *p* = 0.002 in the placebo group) [133]. But, similar to the results of the phase I clinical trials, an elevation in the TG level was observed, reaching 11–13%. To mitigate the rise in the TG level and simultaneously exploit the positive impact of ACC inhibitors on steatosis, several strategies are being evaluated, including (1) the addition of fibrates, which was mentioned earlier [110], and (2) combined therapy with another experimental approach for DGAT2 inhibition (MIRNA study); but those results are expected at the beginning of 2024 [134].

### 4.7. Fatty-acid Synthase Inhibitors

Fatty-acid synthase (FASN) is responsible for endogenous lipogenesis. Owing to its action, cells are exposed to an abundance of palmitate. This leads not only to the accumulation of lipid droplets in hepatocytes but also to the activation of stellate cells, leading to fibrosis. Therefore, the pharmacological inhibition of FASN seems to be a promising therapeutic target [135]. As a result, both lipid load and fibrosis might be inhibited, which are essential pathological pathways in the development of MASH. In a phase I clinical trial, the FASN inhibitor (TVB-2640) effectively reduced de novo lipogenesis in the liver, which was accompanied by a marginally significant ALT reduction (15.8 ± 8.4%; *p* = 0.05) [136]. The recent results of a phase II clinical trial of TVB-2640 (denifanstat) in patients with MASH (FASCINATE-1), show even more promising findings [137]. In a group of 99 subjects treated for 3 months, a significant reduction in liver steatosis was noted using MRI-PDFF in a group receiving a 50 mg dose (28.1 + 28%; *p* = 0.001). The result was accompanied by improvements in LFTs—ALT dropped by 22.3% (*p* < 0.005). The therapy was well-tolerated with minor adverse events. Although promising, those results are preliminary in a phase II clinical trial consisting of a relatively small sample size. Nevertheless, we are expecting new clinical data from an ongoing study with a larger study population and prolonged observation up to 52 weeks [138].

### 4.8. Diacylglycerol Acyltransferase Inhibitors

A reduction in endogenous lipid synthesis is a promising therapeutic strategy in MAFLD treatment. The final stage in the synthesis of triglycerides is mediated by diacylglycerol acyltransferases (DGATs). The inhibition of these enzymes, especially DGAT2, shows promising results. Ervogastat (PF-06865571) has been studied in phase I and phase II clinical trials. In patients with MAFLD, a 6-week treatment resulted in a −35.4% (−47.4, −20.7; *p* = 0.0007) reduction in liver fat, as determined using MRI-PDFF [139]. Additionally, a trend toward improvement in LFT tests was observed. These results, though preliminary, are promising. We are eagerly awaiting the results of an ongoing, larger multicenter study (MIRNA) on the effects of DGAT2 inhibition (with or without the ACC inhibitor—clesacostat) in patients with MASH. The observation time is extended to 48 weeks [140]. A total of 258 patients are included in the study, which should conclude by the end of February 2024 [134]. Another option to reduce the influence of DGAT2 is to reduce its expression. This may be achieved using antisense therapy. ION224 is a ligand-conjugated antisense compound that is currently under development in a phase II clinical trial on 160 patients with MASH, for which the completion is expected in March 2024 [141]. At this time, DGAT2 inhibition is a capable, but still experimental, therapy.

### 4.9. Fibroblast Growth Factor 21

The connection between the lipid overload and progression of MAFLD is also reflected in the experimental usage of fibroblast growth factor 21 (FGF21). FGF21 participates substantially in lipid and glucose metabolism, and it has been suggested that FGF21 may improve the course of MAFLD [142]. A pegylated FGF21 (pegozafermin), in a phase I/IIa clinical trial, showed an acceptable safety profile and efficacy [143], and its efficacy was confirmed in a larger 24-week phase II clinical trial (ENLIVEN) [144]. The study was primarily focused on the fibrosis assessed in liver biopsy specimens and showed significant improvements in MASH. Additionally, it greatly reduced the liver fat content up to 41.9 + 5.6% (vs. 5.0 + 5.2% in the placebo group), as assessed using MRI-PDFF, and a concomitant ALT reduction by up to 31.8 + 5.4%. These promising results warrant further long-term exploration in a phase III clinical trial in this indication. Currently, pegozafermin is being evaluated in a phase III clinical trial but in patients with severe hypertriglyceridemia [145]. There are several other FGF21 analogs that are currently being investigated. Efruxifermin showed promising results in phase II trials [146]; nevertheless, despite meeting the treatment goals in the FALCON program, another long-acting FGF21-based drug—pegbelfermin, has been suspended from further development owing to the sponsor’s decision [147].

## 5. The Role of Ursodeoxycholic Acid in Metabolic-Associated Fatty Liver Disease

UDCA, an agonist of the farnesoid X receptor, has been explored as a promising treatment option for MAFLD. It is a hydrophilic bile acid occurring naturally in the body, but its role still remains uncertain, as was shown in a meta-analysis conducted by Zhang et al. [148]. The meta-analysis demonstrated that UDCA treatment resulted in a significant reduction in ALT levels (*p* = 0.007), especially in the European population (*p* = 0.003), aged over 50 years old (*p* = 0.001). The effectiveness of UDCA in the treatment of NAFLD was more pronounced at longer treatment durations (*p* = 0.008).

Another meta-analysis performed by Lin et al. [149], involving 655 patients in eight studies, confirmed the significant reductions in ALT and GGT levels after UDCA treatment but did not show any notable impact on the liver histology.

Although the undeniable effect of UDCA is the normalization of the liver function parameters (ALT and GGT), its impact on hard endpoints, such as liver fibrosis and liver histology, has not been fully confirmed. Further research is needed to thoroughly assess its potential effects in these areas.

## 6. The Role of the Microbiota in the Pathogenesis and Management of Metabolic-Associated Fatty Liver Disease

The role of the gut microbiota in the pathogenesis and management of many disorders (including MALFD) has been a subject of growing interest. The alterations in the composition and function of the gut microbiota (described as dysbiosis) may contribute to the development and progression of MAFLD. Dysbiosis may lead to increased intestinal permeability, resulting in the translocation of microbial products, such as lipopolysaccharides (LPS), into the bloodstream. This process triggers inflammation and oxidative stress in the liver by, among other pathways, increasing the level of circulating TNF-α (Tumor Necrosing Factor alpha), IL(interleukin)-1, and IL-6, thereby promoting the development of hepatic steatosis and progression of MAFLD [150,151] The involvement of the microbiota in the development of MAFLD has led to the concept of using probiotics, prebiotics, and synbiotics (a combination of probiotics and prebiotics) in the treatment of MAFLD. The modulation of the gut composition through the use of specific strains of probiotic bacteria may potentially help to restore balance and improve liver function in MAFLD [152,153]. Prebiotics (such as fructans, galacto-oligosaccharides, starches and glucose-derived oligosaccharides, and peptic and non-carbohydrate oligosaccharides) are the substances that selectively stimulate the growth and activity of beneficial gut bacteria and promote their beneficial effects [154].

In the last few years, following the plethora of publications in the fields of the microbiota and liver disorders, a few narrative and systematic reviews have been published. Capri et al. [152] presented results of 13 studies, mostly double-blinded, including 947 patients diagnosed with MASH or NAFLD (aged 18−80 years), in which probiotics (six studies), prebiotics (three studies), and synbiotics (four studies) were used. The gut microbiota intervention resulted in an improvement in the markers of inflammation, such as LPS, TNF-α, and IL-6, as well as LFTs. The other parameters, like the levels of lipids, BMI, body weight, waist circumference, insulin sensitivity (as measured using HOMA-IR), fasting blood glucose, and vaspine level, also showed improvement. The liver scores, such as the FLI (fatty-liver index) and steatosis, as well as the fibrosis and NAFLD activity scores decreased.

A meta-analysis of clinical trials involving 1403 patients, investigating the relationship between NAFLD and probiotic supplementation, published by Huang et al. in 2022 [155], revealed that probiotic supplementation improved liver injury in NAFLD patients. A significant reduction in the aminotransferase activity was observed both for ALT (IU/L) [MD (−7.25); 95% CI: (−10.11)–(−4.39), *p* = 0.00001] and AST (IU/L) [MD (−3.53); 95% CI: (−5.62)–(−1.44), *p* = 0.0009], as well as for GGT (IU/L) [MD (−2.27), 95% CI: (−4.49)–(−0.05), *p* = 0.04]. Furthermore, a significant reduction was observed in the insulin level (μIU/mL) [MD (−1.27); 95% CI: (−2.39)–(−0.15), *p* = 0.003], insulin resistance (HOMA-IR) [MD (−0.61); 95% CI: (−1.02)–(−0.21), *p* = 0.03], and body-mass index (Kg/m^2^) [MD (−0.8) 95% CI: (−1.51)–(−0.08), *p* = 0.03]. Also, markers of systemic inflammation, such as TNF-α (pg/mL) [MD (−3.43); 95% CI: (−6.56)–(−1.71), *p* = 0.03] and CRP (mg/dL) [MD (−1.06); 95% CI (−1.94)–(−0.18), *p* = 0.02], were lower after probiotic supplementation.

Another meta-analysis of twenty-one randomized control trials by Sharpton et al. [153], conducted on a group of 1252 participants, revealed a significant reduction in the ALT activity WMD: (−11.23 IU/L); 95% CI: [(−15.02)—(−7.44)] and liver stiffness measurement—WMD: (−0.70 kPa); 95% CI: [(−1.00)—(−0.40 kPa)], although the analyses showed significant heterogeneity (*I*^2^ = 90.6% and *I*^2^ = 93.4%, respectively) in the obtained results. Probiotic and synbiotic administrations were associated with increased odds of improvement in hepatic steatosis, as evaluated using ultrasonography (OR: 2.40; 95% CI: 1.50, 3.84; *I*^2^ = 22.4%). The reduction in the BMI was also associated with probiotic administration WMD: (−1.84); 95% CI: [(−3.30)—(−0.38); *I*^2^ = 23.6%], but there was no positive correlation with synbiotic administration WMD: (−0.85); 95% CI: [(−2.17), 0.47; *I^2^* = 96.6%]. Microbiota modulation provides a modest impact on LFTs, but the relative safety of such therapies is encouraging in clinical practice.

### 6.1. The Role of Helicobacter pylori in the Pathogenesis and Management of Metabolic-Associated Fatty Liver Disease

It has been postulated that *H. pylori* infection might be involved one of the many factors in the complex pathogenesis of MAFLD. *H. pylori* infection can impact the composition and function of the gut microbiota, insulin resistance, and systemic inflammation. [156] Regarding the gut microbiota, *H. pylori* infection can affect its composition and diversity, potentially influencing the development of MAFLD. However, the precise mechanisms underlying this relationship are not yet fully understood. Moreover, the systemic inflammatory state that contributes to the pathogenesis of MAFLD is associated with an *H. pylori* infection, during which the concentration of pro-inflammatory markers increases, potentially influencing the development of MAFLD in a yet-unclear manner [156].

A meta-analysis by Xu et al. based on 34 eligible studies (27 cross-sectional, three case-control, and four cohort studies) revealed a weak association between *H. pylori* infection and NAFLD [157]. Some studies have shown a positive association between *H. pylori* infection and markers of inflammation in individuals with MAFLD. Nevertheless, *H. pylori* is the primary factor behind chronic gastritis and can lead to potential grave duodenal peptic ulcer disease, as well as gastric cancer or gastric mucosa-associated lymphoid-tissue lymphoma. Other diseases, such as unexplained iron-deficiency anemia, vitamin B12 deficiency, and idiopathic thrombocytopenic purpura, can be related to *H. pylori* infection as well. There are also other extra-gastric manifestations of *H. pylori* infection, such as neurodegenerative and cardiovascular diseases, ischemic heart disease, diabetes mellitus, metabolic syndrome, and liver diseases, including nonalcoholic fatty liver disease [158]. The extra-gastric manifestations have been connected to chronic and subclinical systemic inflammation. The data mostly originate from observational studies and are still limited and inconsistent and, therefore, remain inconclusive.

Another factor contributing to the development of MAFLD—insulin resistance, has been shown to be associated with *H. pylori* infection. Some studies suggest a relationship between the eradication of *H. pylori* and an improvement (reduction) in insulin resistance [159,160,161]. However, other studies have not demonstrated this association [162].

Despite these promising findings, it is very important to note that research on the participation of dysbiosis in MAFLD is still evolving. Further studies are needed to establish the precise mechanism as well as optimal strategies for exploiting the microbiota for therapeutic use. One of the proposed mechanisms by which prebiotics may act on MAFLD is through the modulation of the PI3K-AKT/mTOR/AMPK (phosphatidylinositol 3 kinase-protein kinase B/mammalian target of rapamycin/adenosine 5’-monophosphate (AMP)-activated protein kinase) pathway [163], while others highlight the impacts of prebiotics on the lipid profile, including reduced lipid accumulation in the liver (due to the downregulation of lipogenesis and upregulation of fatty-acid oxidation), the recovery of the gut microbiota composition, and enhanced intestinal integrity [164].

### 6.2. Modification of Microbiota

Antibiotics can eliminate unfavorable microbiota, and their efficacy has been confirmed in several liver diseases [165]. For treating cirrhosis and encephalopathy, as well as spontaneous bacterial peritonitis, fluoroquinolones (norfloxacin and ciprofloxacin), third-generation cephalosporins (ceftriaxone and cefotaxime), and trimethoprim–sulfamethoxazole are recommended; and neomycin, metronidazole, polymyxin B, and rifaximin have been used. β-Lactam/β-lactamase inhibitor combinations (BLBLIs) and carbapenems are recommended as the first choice in empirical treatments. Rifaximin, as a eubiotic, has a potential effect on the liver by modulating the bowels’ microbiota. Gangarapu et al. [166] found that the short-term administration of antibiotics improved clinical symptoms in NAFLD/MASH patients by lowering circulating endotoxin and IL-10 levels (0.9 ± 0.34 vs. 0.8 ± 0.13 EU/mL, *p* = 0.03 and 4.08 ± 0.9 vs. 3.73 ± 0.7 pg/mL, *p* = 0.006, respectively) as well as LFTs (AST: 50.4 ± 39 vs. 33 ± 14 IU/L, *p* = 0.01; ALT: 72 ± 48 vs. 45.2 ± 26.3 IU/L, *p* = 0.0001, GGT: 52 ± 33 vs. 41.2 ± 21.1 IU/L, *p* = 0.02). Similarly, a study by Abdal-Razik et al. [167] demonstrated a significant decrease in transaminases and transferase activity [ALT: 64.6 ± 34.2 vs. 38.2 ± 29.2 IU/L, *p* = 0.017; AST: 66.5 ± 42.5 vs. 40.1 ± 20.1 IU/L, *p* = 0.042; GGT: 56.7 ± 31.6 vs. 34.8 ± 28.6 IU/L, *p* = 0.046]; endotoxins; and IL-10, TNF-alpha, and IL-6 levels [0.82 ± 0.22 vs. 0.7 ± 0.09 EU/mL, *p* = 0.001; 40.6 ± 18.78 vs. 58 ± 23.16 pg/mL, *p* = 0.009; 19.18 ± 9.29 vs. 13.34 ± 8.31 pg/mL, *p* = 0.045; 8.34 ± 1.8 vs. 6.67 ± 1.1 pg/mL, *p* = 0.004, respectively]; and the NAFLD Liver Fat Score [from (−0.6) to (−0.2) vs. from (−0.7) to (−0.4) (*p* = 0.034)], following the administration of rifaximin for 6 months but not after 1 month of therapy. The obtained results may be associated with the short duration of the therapy or low drug dose, as found by Ponziani et al. [168]. Moreover, it is important to acknowledge that rifaximin, as a eubiotic, can, indeed, influence the gut microbiota by promoting the growth of beneficial bacteria, particularly *Bifidobacteria* and *Lactobacilli* spp. However, it is crucial to consider that although short-term antibiotic therapy may lead to positive effects, prolonged and systemic antibiotic use can disrupt the balance of gut microbiota, leading to dysbiosis.

In summary, the results obtained from these studies may be linked to the treatment duration, and the effectiveness of rifaximin for treating MASH could have been influenced by factors such as low drug dosages and small sample sizes. Additionally, although several antibiotics can promote the growth of beneficial gut bacteria, the long-term and systemic use of antibiotics should be approached with caution to avoid the potential risk of gut dysbiosis [169].

## 7. Antioxidants

### Vitamin E (Alpha-Tocopherol)

Current international guidelines for the management of liver diseases (e.g., AASLD, AACE, the European Association for the Study of Diabetes (EASD), and the European Association for the Study of Obesity (EASO)) indicate the use of vitamin E (alpha-tocopherol) in the therapy of MASH in patients without diabetes [7]. Lipid accumulation in hepatocytes leads to lipotoxicity and increases the level of oxidative stress, which results in liver injury and inflammation [170]. As a redox scavenger, vitamin E may prevent the damage caused by excessive oxidative stress [171]. The off-label use of vitamin E is connected with improvements in LFTs [172]. The current recommendations on the use of vitamin E in NAFLD treatment are based on the results presented in RCTs concerning patients suffering from MASH with or without type 2 diabetes mellitus (DMT2). In the PIVENS trial, the antioxidative properties of vitamin E appeared to have a positive effect on the liver histology both in reducing the NAFLD activity score (NAS) and resolving MASH [72,173]. In patients with MASH and concomitant DMT2, such changes were not observed [76].

Positive effects of vitamin E on rodent models of MAFLD have been noted [174,175]. Also, some RCTs conducted in adult human patients with MAFLD indicated improvement in the level of liver enzymes [ALT: −7.37 IU/L, 95% CI: (−10.11)–(−4.64); AST: (−5.71) IU/L, 95% CI: (−9.49)–(−1.93)] and histological features (e.g., fibrosis score, with a mean difference of −0.26, 95% CI: (−0.47)–(−0.04) (*I*^2^ = 0%, *p* = 0.02)] [176,177] after therapy with vitamin E. Nevertheless, owing to the inconsistency of the data, to date, there are still a lack of strong recommendations for the use of vitamin E in the management of earlier stages of MAFLD [7].

## 8. Interactions between the Endocrine System and NAFLD

The endocrine system participates, notably, in lipid and glucose metabolisms [178]. For many years, the course of hypothyroidism, hypogonadism, hypopituitarism, and hypercortisolism has been associated with the accumulation of liver fat. Therefore, screening (clinical and/or laboratory) toward endocrine disorders in patients with NAFLD should be introduced on a regular basis to exclude potential reversible causes.

### 8.1. Thyroid Hormones

As mentioned earlier (Section 1), hypothyroidism is an important risk factor for MAFLD [179]. The use of thyroid hormones to induce fat loss in the liver might be a promising therapeutic option, but the burden of side effects precludes their use in native form. However, it might be feasible to use specific agonists of thyroid receptor beta (TRb), which selectively affect liver metabolism while omitting most cardiac effects (e.g., tachycardia) resulting from TRa activation. Currently, there are several compounds under evaluation in the treatment of NAFLD. Resmetirom halved liver fat content, as assessed using MRI-PDFF, in patients with MASH during 36 weeks of treatment with an 80 mg daily dose of the drug [180]. This was accompanied by a significant reduction in ALT [11.0 ± 6.8 vs. (−15.4) ± 4.7 IU/L; *p* = 0.0019] and AST [3.6 ± 2.8 vs. (−7.4) ± 1.9 IU/L; *p* = 0.0016] activities at week 36. The 72-week open-label extension trial also showed resmetirom’s efficacy and sufficient tolerability. Interestingly, although the fat content, as assessed using MRI-PDFF, was significantly reduced, the CAP measurement obtained during Fibroscan remained unaffected [181]. The results of a phase 3 clinical trial evaluating the safety of resmetirom in NAFLD treatment are expected in 2024 [182].

### 8.2. Testosterone

Low testosterone levels (<346 ng/dL) in males have been linked with noninvasive markers of liver steatosis [183]. Long-term observations strongly suggest improvements in liver steatosis in hypogonadal men during testosterone replacement therapy. A 12-year follow-up showed that maintaining the optimal testosterone level with testosterone undecanoate resulted in a significant improvement in the Fatty Liver Index (from 83.6 ± 12.08 to 66.91 ± 19.38; *p* < 0.0001) and GGT (from 39.31 ± 11.62 to 28.95 ± 7.57 IU/L; *p* < 0.0005) but without a significant change in the ALT or AST activity [184]. Therefore, testosterone-replacement therapy is improving the metabolic status of males with hypogonadism. Conversely, testosterone or anabolic androgenic steroids may lead to liver damage in men with normal testosterone levels [185].

### 8.3. Estradiol

Estrogens play an important role in the metabolism of lipids [186]. Estradiol deficiency, which occurs after menopause, coincides with an increased propensity to develop NAFLD [187]. The loss of the protective effects of estrogens is also reflected by a reduction in the sex-hormone binding protein (SHBG), which leads to a relative “excess” in androgen levels [188]. Estradiol therapy might be beneficial for the course of MAFLD after menopause, but it is generally used with progestins, which may alleviate the benefits [189]. Additionally, owing to common contraindications (e.g., thromboembolic events and breast cancer) such a therapy cannot be recommended universally. Therefore, in postmenopausal women, estrogen/progestin-replacement therapy may improve the course of MAFLD; but, currently, MAFLD should not be the sole indication for such a therapy [190]. On the other hand, younger patients (<40 years of age) affected by surgical hypogonadism have an increased risk for developing MAFLD by 50% [191]. In this group of patients, hormone-replacement therapy is generally advisable.

### 8.4. Growth Hormone

Growth hormone (GH) has a multidimensional influence on the human metabolism, including that of the liver. Both deficiency and excess may lead to several pathologies [192]. GH deficiency is commonly associated with liver steatosis, and the risk of NAFLD development in such patients is nearly two-fold OR = 1.85; 95% CI: [(1.05–3.28); *p* = 0.03] [193]. Patients with NAFLD tend to have a reduced peak GH secretory response (9.2 ± 6.4 vs. 15.4 ± 11.2 ng/mL; *p* = 0.001), and higher IGF-1 (Insulin-like Growth Factor-1) levels are associated with less-advanced FIB-4 scores [194]. A small study on young adults with NAFLD and without a definite GH deficiency, but with suboptimal IGF-1 levels, has not shown any significant impact on ALT, AST, and GGT levels. Nevertheless, during the 24-week observation, a trend toward a reduction in liver fat content (−3.3%); 95% CI: [(−7.8%)–1.2%; *p* = 0.14)] was noted [195]. The authors concluded that GH therapy may have a beneficial impact in obese NAFLD patients, but further studies on a larger population of patients are necessary to explore this hypothesis.

## 9. Other Therapies

### 9.1. Xanthine Oxidase Inhibitors

A positive correlation between hyperuricemia and the incidence of MAFLD underlies the research concerning the probably beneficial effects of therapy with xanthine oxidase inhibitors (allopurinol and febuxostat) on liver functions [196].

In the mouse model of MASH, both xanthine oxidase inhibitors appeared to alleviate hepatic steatosis and fibrosis [197,198]. In a pilot interventional study with febuxostat in patients with MAFLD, serum levels of ALT [before: 73.0 (69.8–117.8); after: 70.5 (57.5–94.5) IU/L, *p* = 0.040] and AST [before: 50.5 (40.8–69.8); after: 44.5 (34.8–60.8) IU/L, *p* = 0.018] were significantly decreased, and hepatic steatosis, as confirmed by conducting a histopathological examination, was improved [197]. The results of an ongoing interventional randomized clinical trial comparing the influences of allopurinol and febuxostat on MAFLD (NCT05474560) may clarify the potential use of xanthine oxidase inhibitors for the therapy of liver diseases in the future [199]. Currently, there are no recommendations on the use of xanthine oxidase inhibitors in MAFLD treatment.

### 9.2. Lubiprostone

A gut–liver axis dysfunction turned out to be one of the pathological mechanisms responsible for the progression of MAFLD. Thus, disturbances in the intestinal permeability and dysbiosis have become potential targets in the management of MAFLD [200].

Lubiprostone, an oral metabolite of prostaglandin E_1_, which increases intestinal fluid secretions by promoting the intraluminal chloride-anion efflux, was originally implemented for the therapy of idiopathic constipation and irritable bowel syndrome with constipation [201]. In mouse models, it appeared to improve plasma hepatic injury markers and liver steatosis in MAFLD [202]. In an RCT of 150 patients with constipation and MAFLD (the LUBIPRONE study), after 12 weeks of therapy with lubiprostone, the ALT levels were significantly lower than those in the placebo group MD: (−15) IU/L; 95% CI: [from (−23) to (−6), *p* = 0.0007)]. A significant improvement was also observed in the levels of AST MD: (−9) IU/L; 95% CI: [from (−15) to (−6), *p* = 0.006] and GGT MD: (−15) IU/L; 95% CI: [from (−26) to (−6), *p* = 0.01]. The liver fat content, as assessed using MRI-PDFF, and liver stiffness, as assessed using vibration-controlled transient elastography, also improved [203]. The results of an ongoing RCT with 100 patients (NCT05768334) suffering from MAFLD might determine the future usefulness of drugs that target the gut–liver axis in the therapy of MAFLD [204].

### 9.3. Pentoxifylline

Considering the multifactorial pathogenesis of MAFLD, which includes inflammation, drugs with potential anti-inflammatory effects, such as pentoxifylline, were evaluated for use in the aforementioned liver dysfunction. As a nonspecific phosphodiesterase 4 (PDE-4) inhibitor with the ability to decrease the transcription of TNFa, which is considered as one of the main proinflammatory agents responsible for the degradation of hepatocytes, the effects of therapy with pentoxifylline were examined both in rodents and humans [205,206]. Although early randomized trials in patients with MAFLD confirmed significant improvements in the ALT [WMD: (−13.64) IU/L, 95% CI: (−19.61)–(−7.66), *p* < 0.00001] and AST [WMD: (−9.70) IU/L, 95% CI: (−15.24)–(−4.16], *p* = 0.0006] levels and NAFLD activity score (NAS) [WMD: (−1.16), 95% CI: (−1.51)–(−0.81), *p* < 0.00001] and the regression of lobular inflammation in comparison with the placebo group, further data concerning the positive effects of pentoxifylline on liver functions were inconsistent and, therefore, did not allow the implementation of this drug for the therapy of MAFLD [207,208,209].

### 9.4. Other Drugs

The data from selected meta-analyses might suggest that several drugs commonly used in patients with cardiovascular diseases with concomitant MAFLD (e.g., losartan and acetylsalicylic acid) have positive impacts on hepatic enzyme levels or protect against progression to advanced fibrosis [210,211]. Nevertheless, currently, the level of evidence is underwhelming.

## 10. Functional Foods

### 10.1. Curcumin

Curcumin is the most common curcuminoid found in turmeric, which is widely used in natural medicine and has a pleiotropic effect [212]. It has been shown that curcumin improves glucose and lipid metabolisms, reduces blood pressure, has anti-inflammatory and antioxidant effects, and has positive effects on fat metabolism and weight loss [212]. It is worth mentioning that the lipid-lowering and anti-inflammatory properties of curcumin have been noticed and included in the International Lipid Expert Panel (ILEP) position paper [213,214] and the guidelines of the Polish Lipid Association (PoLA) [215] (Table 1).

The impressive efficiency of curcumin in the prevention of metabolic disorders is highlighted by a meta-analysis conducted by Ashtary-Larky et al., which included the results of nine randomized clinical trials (n = 510 participants). In the analyzed studies, nano-curcumin was used at a dose of 40–120 mg/day for a period of 6–12 weeks [216]. The results of this important meta-analysis are summarized in Table 2.

The properties of curcumin mean that it can have a beneficial effect on the progression of MAFLD. In a meta-analysis of 16 randomized clinical trials conducted by Ngu et al., including 1028 patients with MAFLD, the effect of curcumin supplementation on various metabolic parameters was assessed. Curcumin was shown to reduce the severity of MAFLD (RR = 3.52; 95% CI: 1.27–9.72) and increase liver steatosis resolution (RR = 3.96; 95% CI: 1.54–10.17). Moreover, curcumin was found to reduce the concentrations of AST [MD = (−4.00); 95% CI: (−5.72)–(−2.28) and ALT [MD = (−7.02); 95% CI: (−9.83)–(−4.20)] [217]. The beneficial effect of curcumin on the course of MAFLD was also confirmed in the meta-analysis of 16 randomized clinical trials conducted by Lukkunaprasit et al. It was shown that the use of curcumin by patients with MAFLD was associated with a decrease in the concentration of AST [MD = (−3.90); 95% CI: (−5.97)–(−1.82)], a decrease in ALT [MD = (−5.61); 95% CI: (−9.37)–(−1.85)], an increase in the resolution of hepatic steatosis (as measured using ultrasonography) (MD = 3.53; 95% CI: 2.01–6.22), and a reduced fasting blood sugar, body-mass index, and total cholesterol concentration [218].

The results of these meta-analyses of randomized clinical trials indicate that curcumin may be important in supporting the treatment of patients with MAFLD. However, keep in mind that curcumin from the diet is poorly bioavailable and that you should choose supplements containing curcumin with increased bioavailability (e.g., nano-curcumin formulations [212].

### 10.2. Coffee

Coffee is the most-consumed beverage in the world after water and tea [219]. Coffee contains over 1000 chemical compounds, and the most important include caffeine, chlorogenic acid, trigonelline, caffeic acid, ferulic acid, and melanoidins, as well as kahweol and cafestol [219]. Coffee is characterized by a multidirectional effect. It has been shown that the regular consumption of moderate amounts of coffee may be associated with anti-inflammatory and antioxidant effects [220], antidiabetic effects [221], and antihypertensive effects [222], as well as improving the function of the vascular endothelium [222]. The beneficial effect of coffee on health is confirmed by the fact that the regular consumption of moderate amounts reduces the risk of death from any cause [223].

The described pleiotropic action mechanisms of coffee make it the subject of research on its effects on liver functions. A systematic review of the literature, conducted by Sewter et al., showed that coffee consumption is inversely associated with the severity of hepatic fibrosis in individuals with MAFLD [224]. A meta-analysis of 11 observational studies, conducted by Hayat et al., showed a significantly reduced risk of liver fibrosis in those who drank coffee compared to those who did not drink coffee among MAFLD patients (RR = 0.68; 95% CI: 0.68–0.79) [225]. This beneficial effect of coffee was also confirmed in a meta-analysis of five studies conducted by Kositamongkol et al. It was shown that patients with MAFLD who consumed coffee had a lower probability of liver fibrosis (OR = 0.67; 95% CI: 0.55–0.80) [226]. It is worth mentioning that patients with MAFLD who consume coffee have a lower risk of death from liver cirrhosis (RR = 0.55; 95% CI: 0.35–0.74), as shown in a meta-analysis of nine studies conducted by Kennedy et al. [227].

The results of these studies and their meta-analyses indicate that coffee consumption, by patients with MAFLD, may slow the progression of MAFLD to cirrhosis.

From a practical point of view, patients with MAFLD who declare a desire to consume coffee should be recommended coffee brewed with a paper filter. This is because unfiltered coffee may have a hyperlipemic effect [228], which, in the case of MAFLD, is not a desirable effect. Filtering coffee reduces the content of kahweol and cafestol, i.e., compounds with a hyperlipidemic effect [219] and, thus, eliminates this potentially adverse effect for patients with MAFLD [228].

### 10.3. Resveratrol

Resveratrol is a polyphenolic derivative of stilbene and is commonly found in nuts, grapes (red wines), blueberries, tomato skins, and cocoa [229]. Resveratrol is characterized by, among other things, antioxidant, anti-inflammatory, cardioprotective, and antidiabetic effects [229].

Despite the several beneficial properties of resveratrol, the research results do not indicate its significant role in supporting the treatment of MAFLD. In a meta-analysis of four randomized clinical trials, conducted by Zhang et al., covering 156 patients with MAFLD, it was shown that the use of resveratrol did not affect the body weight; BMI; systolic or diastolic blood pressure; tissue sensitivity to insulin (HOMA-IR); or ALT, AST, GGT, bilirubin, or TNF-alpha levels [230]. Similar results were obtained in a meta-analysis of seven randomized clinical trials, conducted by Jakubczyk et al., including 302 patients with MAFLD. Resveratrol was administered daily over periods between 56 and 180 days in doses ranging from 500 mg to 3000 mg per day. No effect of resveratrol supplementation (irrespective of the dose and duration of the intervention) was found on the AST, body weight, BMI, WC, glucose or insulin concentration, total cholesterol, TG, LDL, HDL, or systolic or diastolic blood pressure [231].

The results of the meta-analyses of randomized clinical trials indicate that the use of resveratrol in patients with MAFLD does not offer any clinical benefits.

### 10.4. Vitamin D

Approximately 20% of the total vitamin D in the human body comes from the diet, while 80% comes from endogenous synthesis in the skin [232]. It is a steroid hormone involved in the absorption of calcium and phosphate in the small intestine. Several other mechanisms of action are also attributed to vitamin D, including reducing the activity of the renin–angiotensin system, antioxidant and anti-inflammatory effects, and regulating the functions of adipocytes and pancreatic beta cells [233].

In a meta-analysis of nine studies, conducted by Eliades et al., it was shown that patients with MAFLD have a 26% higher risk of vitamin D deficiency compared to healthy people (OR = 1.26; 95% CI: 1.17–1.35) [234]. Therefore, it is important whether vitamin D supplementation in patients with MAFLD can provide clinical benefits. In a meta-analysis of seven studies, by Sindhughos et al., including 735 patients with MAFLD, it was shown that vitamin D supplementation was associated with improved tissue sensitivity to insulin [HOMA-IR: MD = (−1.06); 95% CI: (−1.66)–(−0.45)] and a decrease in AST [MD = (−4.44); 95% CI: (−8.24)–(−0.65)] [235]. Consistent results were obtained in a meta-analysis of 16 randomized clinical trials, conducted by Rezaei et al., including patients with NAFLD. It was shown that vitamin D supplementation was associated with an increase in HDL-C (*p* = 0.008) and decreases in the body weight (*p* = 0.007), body-mass index (*p* = 0.002), waist circumstance (WC) (*p* = 0.02), serum ALT (*p* = 0.01), fasting blood sugar (*p* = 0.01), and tissue resistance to insulin (HOMA-IR; *p* = 0.004) [236].

The results of the meta-analyses of randomized trials indicate that vitamin D supplementation may contribute to the improvement in the metabolic profiles of patients with MAFLD.

### 10.5. Omega-3 Fatty Acids

Omega-3 fatty acids (ω-3 PUFA) are naturally present in animals (fish, krill, eggs, and squid) and plants (algae, flaxseeds, walnuts, edible seeds, and clary sage) [237]. The mechanisms of action of omega-3 fatty acids, which may have a beneficial effect on the course of MAFLD, include anti-inflammatory and antioxidant effects, a reduction in TG production, a reduction in fat accumulation in the liver, and an improvement in the composition of the intestinal microbiota [26]. In a meta-analysis of 18 randomized clinical trials, by Yan et al., including 1424 patients with MAFLD, omega-3 fatty acids were associated with improvements in fat accumulation in the liver (RR = 1.56; 95% CI: 1.23–1.97) and ALT [SMD = (−0.50); 95% CI: (−0.88)–(−0.11)], decreases in AST [SMD = (−0.54); 95% CI: (−1.04)–(−0.05)] and GGT, [SMD = (−0.48); 95% CI: (−0.64)–(−0.31)], and improved tissue insulin sensitivity [HOMA-IR; WMD = (−0.40); 95% CI: (−0.58)–(−0.22)] [238]. Consistent results were obtained in a meta-analysis of six randomized clinical trials, by Moore et al., including 362 MAFLD patients, which showed that omega 3 fatty-acid supplementation was associated with a reduction in ALT levels [MD = (−8.04); 95% CI: −14.70 to −1.38)] [239].

The results of the meta-analyses of randomized clinical trials indicate that omega-3 fatty-acid supplementation may improve the metabolic profiles and liver functions of patients with MAFLD.

### 10.6. Silymarin

Silymarin is a flavone derivative obtained from milk-thistle fruit. By stabilizing the membranes of liver cells, it has a protective effect on the liver parenchyma. It has stabilizing, regenerating, and protective effects on the membranes of liver cells; weakly relaxes smooth muscles; stimulates the production and secretion of bile; and is anti-inflammatory and strongly detoxifying [240]. In a meta-analysis of eight randomized clinical trials, conducted by Kalopitas et al., including 622 patients with MAFLD, it was found that silymarin supplementation was associated with a reduction in ALT levels [MD = (−14.86); 95% CI: (−19.37)–(10.36)] and AST levels [MD = (−7.11); 95% CI: (−14.16)–(−0.05)]. The small amount of available data does not allow the elucidation of the effect of silymarin on the process of liver fibrosis [241].

The results of the meta-analysis of randomized clinical trials indicate that silymarin supplementation may reduce the activity of liver enzymes in patients with MAFLD.

### 10.7. Garlic

The health benefits of garlic have been known for a long time. Garlic is characterized by anti-inflammatory, antioxidant, anti-aggregation, lipid-lowering, antihypertensive effects, etc. [242]. In a meta-analysis by Yu et al. for 139 patients with MAFLD, garlic supplementation was associated with decreases in ALT levels [MD = (−9.00) 95% CI: (−11.75)–(−6.24)] and AST [MD = (−5.03); 95% CI: (−7.15)–(−2.91)] [243]. The beneficial effect of garlic on the metabolic profiles of patients with MAFLD was also confirmed in the meta-analysis by Rastkar et al. Their meta-analysis included 186 patients with MAFLD, and it showed that garlic supplementation was associated with a significant reduction in the concentrations of ALT, AST, total cholesterol, LDL-C, triglycerides, and fasting glucose. Moreover, the probability of a decrease in hepatic steatosis was 2.75 times lower in the garlic group compared to the placebo group (RR = 2.75; 95% CI: 1.79–4.23) [244].

## 11. Future Perspectives

Owing to the multifaceted pathophysiology of MAFLD, in current clinical trials, new drugs from different groups have been evaluated:farnesoid X receptor agonists—obeticholic acid and tropifexor;a lipogenesis inhibitor—aramchol;a galectin 3 inhibitor—belapectin;an A3 adenosine receptor agonist—namodenoson;a fatty acid—icosabutate;a cyclophilin inhibitor—rencofilstat;a modified bile acid—non-ursodeoxycholic.

In most of these clinical trials, the primary endpoint concerns an improvement in liver fibrosis [245].

## 12. Summary

The pathogenesis of MAFLD is multifactorial, and the pathogenic drivers of MAFLD can serve as therapeutic targets. Those can include: the modulation of food intake, increase in energy expenditure, improvement in adipocyte insulin sensitivity, inhibition of de novo lipogenesis, tapering of oxidative stress, and reduction in inflammation. MAFLD might be considered as a part of the clinical presentation of the metabolic syndrome. For several decades, there have been major improvements in the therapies of diabetes, hyperlipidemia, and hypertension. It seems that drugs affecting incretin receptors, PPAR, and TRb have significant impacts on the surrogate markers of liver steatosis. Based on the available research findings and recommendations from liver disease societies, it appears that some of the most significant factors are the achievement of a normal body weight and the effective treatment of metabolic disorders. Still, there is a lack of a sufficient number of studies indicating which of the cardiometabolic risk factors, included in the definition of MAFLD, most significantly lead(s) to disease progression. Thus, the need for further studies on therapies that will significantly and independently mitigate both the consequences and severity of liver steatosis seems essential. Therefore, alongside the established roles of pioglitazone and vitamin E in MASH treatment, it seems that drugs for weight reduction (including incretin-based medications that also have a glycemic-normalizing effect) and other drugs that affect the cardiometabolic risk will gradually gain importance; hence, the incessant necessity for exploring pathophysiological pathways that would allow a common mechanism of action to be found for steatosis resolution regardless of the type of factor that led to it. Therefore, we are anxiously anticipating the results of ongoing clinical trials not only regarding pharmacotherapy but also more definitive studies on the impact of the TLC and microbiota on the course of MAFLD.

In summary, based on our literature review, it seems that in patients with liver steatosis and concomitant diseases, a recommendation toward the following therapeutic itinerary may be offered:Physical activity and modification of the diet are of the utmost importance;The strengthening of weight loss with pharmacotherapy (preferably with GLP-1 analogs) is advisable;The introduction of functional foods might improve MAFLD;The consideration of the use of pro/prebiotics—modification of microbiota (screening for *H. pylori*);The exclusion of hypothyroidism, hypogonadism, and GH deficiency;In patients with DMT2, a preference toward novel antidiabetic drugs, i.e., SGLT-2i and incretin-based therapies;In patients with lipid disorders, the use of lipid-lowering drugs should be encouraged, while the risk of liver damage seems not to be greater than that in patients without liver steatosis, and the potential beneficial effects on liver functions seem to be concurrent with cardiovascular benefits.

## Figures and Tables

**Figure 1 medicina-59-01789-f001:**
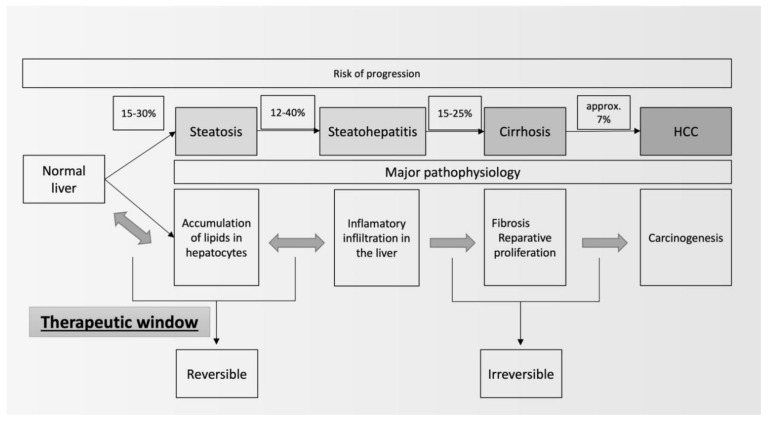
Natural history of MAFLD (modified from [6]).

**Figure 2 medicina-59-01789-f002:**
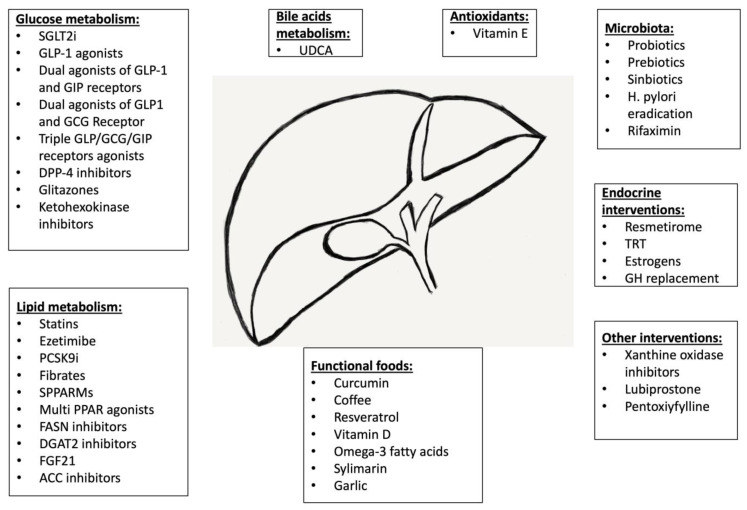
Promising therapeutic options affecting early phases of MAFLD.

**Table 1 medicina-59-01789-t001:** Clinical relevance of curcumin based on existing recommendations: (A) lipid-lowering properties based on the International Lipid Expert Panel (ILEP) position paper [213], (B) the place of curcumin in lipid-lowering therapy based on the Polish guidelines [214], (C) the role of curcumin in managing inflammatory parameters based on the ILEP position paper [215].

**(A)**						
**Class**	**Level**	**Daily Doses**	**Expected Effects on LDL-C**	**Effects on other CV Risk Biomarkers**		**Direct Vascular Effects**
IIa	B	1–3 g	−5%	↓ TG, Lp(a), glucose, HbA_1c_, HOMA index, hs-CRP, TNF-α, and IL-6; ↑ adiponectin and HDL-C		↑ FMD; ↓ PWV
**(B)**						
**Name**	**Recommended Dosage**		**Expected Effects on LDL-C**	**Class of Recommendation**	**Level of Recommendation**	
Curcumin	0.5–3 g		from –5% to –10%	IIa	A	
**(C)**						
**Nutraceuticals**	**Class**		**Level**	**Impact on Markers of Inflammation**		
Curcumin	IIa		B	A significant decrement in serum concentrations of TNF-α (−4.69 pg/mL), IL-6, TGF-β, and MCP-1		

Abbreviations: LDL-C—low-density lipoprotein cholesterol; CV—cardiovascular; TG—triglyceride; Lp(a)—lipoprotein a; HOMA index—Homeostasis Model Assessment of Insulin Resistance; CRP—C-reactive protein; TNF-α—tumor necrosis factor α; IL-6—interleukin 6; HDL-C—high-density lipoprotein cholesterol; FMD—flow-mediated dilation; PWV—pulse wave velocity; TGF-β—transforming growth factor β; MCP-1—monocyte chemoattractant protein-1.

**Table 2 medicina-59-01789-t002:** The effect of nano-curcumin supplementation on the control of cardiovascular parameters (based on information from [216]).

Metabolic Parameter	WMD	95% CI	Comments
**TG (mg/dL)**	−24.87 −27.23	from −37.34 to −12.40; *p* < 0.001 from −43.11 to −11.35; *p* = 0.001	Baseline TG ≥ 150 mg/dL Obese (>30 Kg/m^2^)
**TC (mg/dL)**	−10.90	from −16.40 to −5.39; *p* < 0.001	Baseline TC ≥ 200 mg/dL and obese (>30 Kg/m^2^)
**LDL-C (mg/dL)**	−13.70	from −19.26 to −8.13; *p* < 0.001	Baseline LDL-C ≥ 100 mg/dL and obese (>30 Kg/m^2^)
**HDL-C (mg/dL)**	5.77	from 2.90 to 8.64; *p* < 0.001	Overall effect
**FBG (md/dL)**	−18.14	from −29.31 to −6.97; *p* = 0.001	Overall effect
**Fasting Insulin**	−1.21	from −1.43 to −1.00; *p* < 0.001	Overall effect
**HOMA-IR**	−0.28	from −0.33 to −0.23; *p* < 0.001	Overall effect
**SBP (mmHg)**	−7.09	from −12.98 to −1.20; *p* < 0.001	Overall effect
**CRP (mg/L)**	−1.29	from −2.15 to −0.44; *p* = 0.003	Overall effect
**IL-6 (pg/mL)**	−2.78	from −3.76 to −1.79; *p* < 0.001	Overall effect

Abbreviations: WMD—weighted mean difference; TG—triglyceride; TC—total cholesterol; LDL-C—low-density lipoprotein cholesterol; HDL-C—high-density lipoprotein cholesterol; FBG—fasting blood glucose; HOMA-IR—Homeostasis Model Assessment of Insulin Resistance; SBP—systolic blood pressure; CRP—C-reactive protein; IL-6—interleukin 6.

## Data Availability

All the data have been included in the manuscript.

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
