# Peer review of "Potential Therapeutic Strategies in the Treatment of Metabolic-Associated Fatty Liver Disease"

_medicina, 2023, doi:10.3390/medicina59101789_

Round 1
Reviewer 1 Report
The manuscript is interesting and generally well written. Only minor revisions are required. In particular:
Please kindly remove acronyms from the title and abstract.
Please kindly remove acronyms from the subheading.
The role of Vitamin E should be more elucidated.
Summary section is inadequate and no consensus conclusion is presented.
Moderate editing of English language required
Author Response
Dear Reviewer #1:
Authors are grateful for the review of the manuscript. Below, please find answers for the issues raised. Changes in the manuscript have been marked in red fonts.
The manuscript is interesting and generally well written. Only minor revisions are required. In particular:
Please kindly remove acronyms from the title and abstract.
As proposed by the Reviewer the acronyms from the title and abstract have been removed.
Please kindly remove acronyms from the subheading.
According to recommendation acronyms were removed from subheadings. Definition of acronyms have been moved to following paragraphs of the manuscript.
The role of Vitamin E should be more elucidated.
A section regarding the impact of vitamin E on the resolution of oxidative stress, which participates in the course of MAFLD has been added:
“Current international guidelines for the management of liver diseases [e.g. AASLD, AACE, European Association for the Study of Diabetes (EASD), European Association for the Study of Obesity (EASO)] vote for using vitamin E (alpha-tocopherol) in therapy of MASH in patients without diabetes [7]. Lipid accumulation in hepatocytes lead to lipotoxicity and increases the level of oxidative stress, which result in liver injury and inflammation (165). As a redox scavenger vitamin E may prevent the damage caused by excessive oxidative stress (166). The off-label use of vitamin E is connected with improvements in LFTs (167). The current recommendation on the use of vitamin E in NAFLD are based on the results presented in RCTs concerning patients suffered from MASH with or without type 2 diabetes mellitus (DMT2). In PIVENS trial, Vitamin E with its antioxidative properties appeared to have positive effect on liver histology both in reduction in NAFLD activity score (NAS) and MASH resolution [72,168]. In patients with MASH and concomitant DMT2 such changes were not observed [76]. ”
Summary section is inadequate and no consensus conclusion is presented.
As suggested by the reviewer summary sections has been extended and shaped toward more conlusive recommendations”
“12. Summary
Pathogenesis of MAFLD is multifactorial and the pathogenic drivers of MAFLD can serve as therapeutic targets. Those can include: modulation of food intake, increase in energy expenditure, improve in adipocyte insulin sensitivity, inhibition of de novo lipogenesis, tapering of oxidative stress and reduction of inflammation. MAFLD might be considered as a part of clinical presentation of metabolic syndrome. Since several decades there have been major improvements in the therapy of diabetes, hyperlipidemia, hypertension. It seems that drugs affecting incretin receptors, PPAR and TRb have significant impact on surrogate markers of liver steatosis. Based on available research findings and recommendations from liver disease societies as well, it appears that one of the most significant factors is achieving a normal body weight and effective treatment of metabolic disorders. Still, there is a lack of a sufficient number of studies indicating which of the cardiometabolic risk factors, included in the definition of MAFLD, most significantly leads to disease progression. In light of this, the need for further studies of the therapies that will significantly reduce both the consequences and liver steatosis independently seems essential. Therefore, alongside the established role of pioglitazone and vitamin E in MASH treatment, it seems that drugs for weight reduction (including incretin-based medications that also have a glycemic-normalizing effect) and other drugs affecting cardiometabolic risk will gradually gain importance. Hence, the incessant necessity for exploring pathophysiological pathways that would allow finding a common mechanism of action in steatosis resolution regardless of the type of factor that led to it. Therefore, we are anxiously anticipating the results of ongoing clinical trials not only regarding pharmacotherapy, but also the results of more definitive studies on the impact of TLC and microbiota on the course of MAFLD.
In summary, based on our literature review it seems that in patients with liver steatosis and concomitant diseases, a recommendation toward therapeutic itinerary may be offered:
- Physical activity and modification of diet is of utter importance,
- Strengthening of weight loss with pharmacotherapy (preferably with GLP-1 analogs) is advisable,
- Introduction of functional food might improve MAFLD,
- Consideration on the use of pro/prebiotics - modification of microbiota (screening for pylori),
- Exclusion of hypothyroidism, hypogonadism and GH deficiency,
- In patients with DMT2 preference toward novel antidiabetic drugs i.e., SGLT-2i and incretin-based therapies
- In patients with lipid disorders use of lipid lowering drugs should be encouraged, while the risk of liver damage seems not to be greater than in patients without liver steatosis and potential benefit on liver function seem to be concurrent with cardiovascular benefits.”

Reviewer 2 Report
The manuscript discusses the multifaceted pathophysiology of MAFLD and potential therapeutic interventions. It provides a comprehensive overview of current treatment options and emerging therapies, emphasizing lifestyle modifications, functional foods, and pharmaceutical approaches. However, the paper lacks a critical appraisal of existing clinical evidence and does not sufficiently address limitations, such as small sample sizes in clinical trials. While informative, the paper needs to strengthen its scientific rigor and coherence to be suitable for publication.
Another major concern that must be addressed is that the paper completely ignores the drugs in the pipeline for NAFLD tx – for example, inhibitors of ACC were discussed in 1-2 sentences, FASN – no mention, DGAT – no mention and FGF21 – only in “future” section listed. They must be added to the manuscript.
Author Response
Dear Reviewer #2
Thank you for a thorough review of the submitted manuscript. Authors modified the paper according to the recommendations. Changes in the manuscript are marked in red fonts.
The manuscript discusses the multifaceted pathophysiology of MAFLD and potential therapeutic interventions. It provides a comprehensive overview of current treatment options and emerging therapies, emphasizing lifestyle modifications, functional foods, and pharmaceutical approaches. However, the paper lacks a critical appraisal of existing clinical evidence and does not sufficiently address limitations, such as small sample sizes in clinical trials. While informative, the paper needs to strengthen its scientific rigor and coherence to be suitable for publication.
As suggested by the review the manuscript was supplemented with more critical commentary to the provided data, which generally was put at the end of paragraphs.
Another major concern that must be addressed is that the paper completely ignores the drugs in the pipeline for NAFLD tx – for example, inhibitors of ACC were discussed in 1-2 sentences, FASN – no mention, DGAT – no mention and FGF21 – only in “future” section listed. They must be added to the manuscript.
As advised by the reviewer paragraphs regarding FASN, DGAT and FGF21 have been added to the manuscript (and included in the Fig. 2):
“4.6. Fatty acid synthase inhibitors
Fatty acid synthase (FASN) is responsible for endogenous lipogenesis. Due to its action, cells are exposed to abundance of palmitate. It leads not only to accumulation of lipid droplets in hepatocytes but also to activation of stellate cells leading to fibrosis. Therefore, pharmacological inhibition of FASN seems to be a promising therapeutic target [130]. As a result, both lipid load and fibrosis might be inhibited, which are essential pathological pathways in the development of MASH. Recent results of Phase II clinical trial of TVB-2640 (denifanstat) in patients with MASH (FASCINATE-1) [131]. In a group of 99 subjects treated for 3 months a significant reduction in liver steatosis was noted using MRI-PDFF in a group receiving 50mg dose (28,1+28%; p=0.001). The result was accompanied by improvements in LFTs – ALT dropped by 22.3% (p<0.005). The therapy was well-tolerated with minor adverse events. While promising, those results are preliminary in Phase II clinical trial consisting of relatively small sample size. Nevertheless, we are expecting new clinical data from ongoing study with larger study population and prolonged observation up to 52 weeks [132].
4.7. Diacylglycerol acyltransferase inhibitors
Reduction of endogenous lipid synthesis is a promising therapeutic strategy in MAFLD. The final stage in the synthesis of triglycerides is mediated by diacylglycerol acyltransferases (DGATs). Inhibition of these enzymes, especially DGAT2, shows promising results. Ervogastat (PF-06865571) has been studied in Phase I and Phase II clinical trials. In patients with MAFLD a 6-week treatment resulted in −35.4% (−47.4, −20.7; P=0.0007) reduction in liver fat using MRI-PDFF [133]. Additionally, a trend toward improvement in LFT tests was observed. The results, though preliminary, are promising. We are eagerly awaiting the results of an ongoing, larger multicenter study (MIRNA) on effects of DGAT2 inhibition (with or without ACC inhibitor - clesacostat) in patients with MASH. The observation time is extended to 48 weeks [134]. 258 patients have been included in the study that should conclude by the end of February 2024 [135]. Another option to reduce the influence of DGAT2 is to reduce its expression. This may be obtained by antisense therapy. ION224 is a ligand-conjugated antisense compound that is currently under development in Phase II clinical trial on 160 patients with MASH and its completion is expected in March 2024 [136]. At this moment DGAT2 inhibition is capable but still experimental therapy.
4.8. Fibroblast growth factor 21
The connection between lipid overload and progression of MAFLD is also reflected in the experimental usage of fibroblast growth factor 21 (FGF21). FGF21 participates greatly in the lipid and glucose metabolism and it was suggested that it may improve the course of MAFLD [137]. A pegylated FGF21 (pegozafermin) in a phase I/IIa clinical trial showed acceptable safety profile and efficacy [138] and its efficacy confirmed in larger 24-week phase II clinical trial (ENLIVEN) [139]. The study was primarily focused on the fibrosis assessed in liver biopsy specimen and showed significant improvements in MASH. Additionally, it greatly reduced liver fat content up to 41.9+5.6% (vs. 5.0+5.2% in a placebo group) assessed by MRI-PDFF with a concomitant ALT reduction by up to 31.8+5.4%. Those promising results warrant further long-term exploration in phase III clinical trial in this indication. Currently, pegozafermin is evaluated in a phase III clinical trial, but in patients with severe hypertriglyceridemia [140]. There are several other FGF21 analogs that are currently investigated. Efruxifermin showing promising results in phase II trials [141], Nevertheless, despite meeting treatment goals in FALCON program, another long-acting FGF21 based drug - pegbelfermin, has been suspended in the further development to sponsor decision [142].”

Reviewer 3 Report
Comments for Manuscript
(1) A well written review.
Minor comments:
(2) Authors should replace NASH with the new nomenclature MASH throughout the text.
(3) Authors should add a section on estrogen as it relates to MAFLD/NAFLD, since they have discussed testosterone.
(4) The authors recommended physical activity and modification of diet as one of therapeutic itinerary for liver steatosis and concomitant diseases, but they did not discuss physical activity as it relates to MAFLD/ NAFLD in the manuscript. They should include a brief section on physical activity.
(5) Minor editing of the manuscript is necessary.
(6) A comprehensive list of all abbreviations used in the manuscript must be included before accepting the manuscript.
Minor editing of the manuscript is necessary.
Author Response
Dear Reviewer #3
Authors wish to extend their gratitude for the time spend on the review of the manuscript. Changes to the manuscript have been marked in red fonts.
A well written review.
Minor comments:
Authors should replace NASH with the new nomenclature MASH throughout the text.
As recommended by the reviewer NASH has been substituted with MASH.
Authors should add a section on estrogen as it relates to MAFLD/NAFLD, since they have discussed testosterone.
A paragraph regarding estrogen use, predominantly in the field of hormone replacement therapy has been introduced in the body of manuscript (and incorporated in the Fig.2).
The authors recommended physical activity and modification of diet as one of therapeutic itinerary for liver steatosis and concomitant diseases, but they did not discuss physical activity as it relates to MAFLD/ NAFLD in the manuscript. They should include a brief section on physical activity.
As mentioned by the reviewer a section dealing with the importance of physical exercises has been added:
“Recommendations for increased physical activity have been incorporated in the guidelines of most scientific societies dealing with MAFLD. AASLD [10] recommends that patients with NAFLD should be encouraged to increase their activity level to the possible extent. Individualized prescriptive exercise recommendations may enhance sustainability and offer benefits independent of weight loss. EASL also recommends a progressive increase in aerobic exercise and resistance training [8].
One of the recently published systematic reviews [11], demonstrated that physical activity had a strong association with improvements in inflammation, reduction in steatohepatitis, and fibrosis in experimental models. Furthermore, in human studies, both aerobic and resistance exercise were shown to reduce liver fat and improve insulin resistance and blood lipids, irrespective of weight loss, with aerobic exercises possibly being more effective. This review also showed that resistance training is more achievable for patients with NAFLD who have poor cardiorespiratory fitness (CRF). Meta-analysis by Wang [12] showed that physical activity was associated with minute reductions in LFTs: ALT [SMD = (-0.17 IU/L), 95% CI: (-0.30) - (-0.05)], AST [SMD = (-0.25 IU/L), 95% CI: (-0.38) - (-0.13)], and GGT [SMD = (-0.22 IU/L), 95% CI: (-0.36) - (-0.08). Similar findings were seen in the meta-analysis and meta-regression performed by Xiong et al. [13] indicated that aerobic exercises in patients with NAFLD could significantly reduce activity of ALT [WMD = (-6.14 IU/L), 95%CI: (-10.99) - (-1.29)], AST [WMD = (-5.73 IU/L), 95%CI: (-9.08) - (-2.38)] and BMI [WMD = (-0.85 kg/m2), 95%CI: (-1.19) - (-0.51)]. Additionally, resistance exercises could significantly reduce AST activity [WMD = (-2.58 IU/L), 95%CI: (-4.79) – (-0.36)]) while the high-intensity interval training could significantly reduce ALT activity [WMD = (-6.20 IU/L), 95%CI: (-9.34) – (-3.06)] in patients with NAFLD.”
Minor editing of the manuscript is necessary.
A thorough check of the manuscript have been performed during the revision. Several editorial corrections have been applied and have been marked in red fonts.
A comprehensive list of all abbreviations used in the manuscript must be included before accepting the manuscript.
A complete and updated list of abbreviations has been located at the end of the manuscript prior to the list of references.

Round 2
Reviewer 2 Report
The authors have made revisions that improved the manuscript. However, the authors are recommended to add a similar paragraph on ACC inhibitors. The authors are also recommended to cite the following key papers in the field: PMIDs: 20041406, 24316260, 28768177, and 31630414.
Author Response
Dear Reviewer #2,
Once again thank you for the time spent to review the manuscript. Below please find the changes made in the paper. Current revision is marked in violet fonts (previous round in red fonts).
The authors have made revisions that improved the manuscript. However, the authors are recommended to add a similar paragraph on ACC inhibitors. The authors are also recommended to cite the following key papers in the field: PMIDs: 20041406, 24316260, 28768177, and 31630414.
As recommended by the reviewer a separate paragraph regarding the ACC inhibitors has been added. The suggestion regarding vital references has also been included:
“4.6. Acetyl-CoA carboxylase inhibitors
Cellular lipid overload is a consequence of both exogenous and endogenous lipid excess [130]. Experimental data strongly suggest that patients with MAFLD experience elevated (over 3-fold) de novo fatty acid synthesis [131]. Therefore, the endogenous synthesis seems to be important target for pharmacological approach to MAFLD treatment. One of the options is the inhibition of hepatocyte specific Acetyl-CoA carboxylases (ACC). Initial experiments on MK-4074, a ACC1 and ACC2 inhibitor, led to a remarkable reduction in the liver content that reached 36% (8.6% in placebo group) after 4-week therapy [132]. Unfortunately, probably due to concurrent inhibition of PUFA-3 elongation a rise in TG level was noted (170 vs. 325mg/dL). Further, phase II clinical trials on other ACC inhibitor (firsocostat) show dose dependent reductions in liver steatosis assessed by MRI-PDFF, reaching 28.9 % (vs. 8.4 %; p=0.002 in placebo group) [133]. But similarly, to phase I clinical trials an elevation in TG level was seen reaching 11 - 13 %. In order to mitigate the rise TG and simultaneously exploit the positive impact on steatosis of ACC inhibitors, several strategies are evaluated: (1) addition of fibrates, which was mentioned earlier [110]; (2) combined therapy with another experimental approach with DGAT2 inhibition (MIRNA study), but those results are expected in the beginning of 2024 [134].”
References:
- Fabbrini E, Sullivan S, Klein S. Obesity and nonalcoholic fatty liver disease: biochemical, metabolic, and clinical implications. Hepatol Baltim Md. 2010;51(2):679-689. doi:10.1002/hep.23280
- Lambert JE, Ramos-Roman MA, Browning JD, Parks EJ. Increased de novo lipogenesis is a distinct characteristic of individuals with nonalcoholic fatty liver disease. Gastroenterology. 2014;146(3):726-735. doi:10.1053/j.gastro.2013.11.049
- Kim CW, Addy C, Kusunoki J, et al. Acetyl CoA Carboxylase Inhibition Reduces Hepatic Steatosis but Elevates Plasma Triglycerides in Mice and Humans: A Bedside to Bench Investigation. Cell Metab. 2017;26(2):394-406.e6. doi:10.1016/j.cmet.2017.07.009
and
“Fatty acid synthase (FASN) is responsible for endogenous lipogenesis. Due to its action, cells are exposed to abundance of palmitate. It leads not only to accumulation of lipid droplets in hepatocytes but also to activation of stellate cells leading to fibrosis. Therefore, pharmacological inhibition of FASN seems to be a promising therapeutic target [135]. As a result, both lipid load and fibrosis might be inhibited, which are essential pathological pathways in the development of MASH. In a phase I clinical trial FASN inhibitor (TVB-2640) effectively reduced de novo lipogenesis in the liver, which was accompanied by a marginally significant ALT reduction (15.8 ± 8.4%; p=0.05) [136]”
Reference:
- Syed-Abdul MM, Parks EJ, Gaballah AH, et al. Fatty Acid Synthase Inhibitor TVB-2640 Reduces Hepatic de Novo Lipogenesis in Males With Metabolic Abnormalities. Hepatol Baltim Md. 2020;72(1):103-118. doi:10.1002/hep.31000
